# Attention Overlap Is Responsible for The Entity Missing Problem in Text-to-image Diffusion Models!

**Arash Marioriyad**                                                     *arashmarioriyad@gmail.com*
*Department of Computer Engineering*
*Sharif University of Technology*

**Mohammadali Banayeeanzade**                                            *a.banayeean@gmail.com*
*Department of Computer Engineering*
*Sharif University of Technology*

**Reza Abbasi**                                                          *reza.abbasi.uni@gmail.com*
*Department of Computer Engineering*
*Sharif University of Technology*

**Mohammad Hossein Rohban**                                              *rohban@sharif.edu*
*Department of Computer Engineering*
*Sharif University of Technology*

**Mahdieh Soleymani Baghshah**                                          *soleymani@sharif.edu*
*Department of Computer Engineering*
*Sharif University of Technology*

**Reviewed on OpenReview:** *https://openreview.net/forum?id=Xv3ZrFayIO*

## Abstract

Text-to-image diffusion models such as Stable Diffusion and DALL-E have exhibited impressive capabilities in producing high-quality, diverse, and realistic images based on textual prompts. Nevertheless, a common issue arises where these models encounter difficulties in faithfully generating every entity specified in the prompt, leading to a recognized challenge known as entity missing in visual compositional generation. While previous studies indicated that actively adjusting cross-attention maps during inference could potentially resolve the issue, there has been a lack of systematic investigation into the specific objective function required for this task. In this work, we thoroughly investigate three potential causes of entity missing from the perspective of cross-attention maps: insufficient attention intensity, excessive attention spread, and significant overlap between attention maps of different entities. Through comprehensive empirical analysis, we found that optimizing metrics that quantify the overlap between attention maps of entities is highly effective at mitigating entity missing. We hypothesize that during the denoising process, entity-related tokens engage in a form of competition for attention toward specific regions through the cross-attention mechanism. This competition may result in the attention of a spatial location being divided among multiple tokens, leading to difficulties in accurately generating the entities associated with those tokens. Building on this insight, we propose four overlap-based loss functions that can be used to implicitly manipulate the latent embeddings of the diffusion model during inference: Intersection over union (IoU), center-of-mass (CoM) distance, Kullback–Leibler (KL) divergence, and clustering compactness (CC). Extensive experiments on a diverse set of prompts demonstrate that our proposed training-free methods substantially outperform previous approaches on a range of compositional alignment metrics, including visual question-answering, captioning score, CLIP similarity, and human evaluation. Notably, our method outperforms the best baseline by 9% in human evaluation.

# 1    Introduction

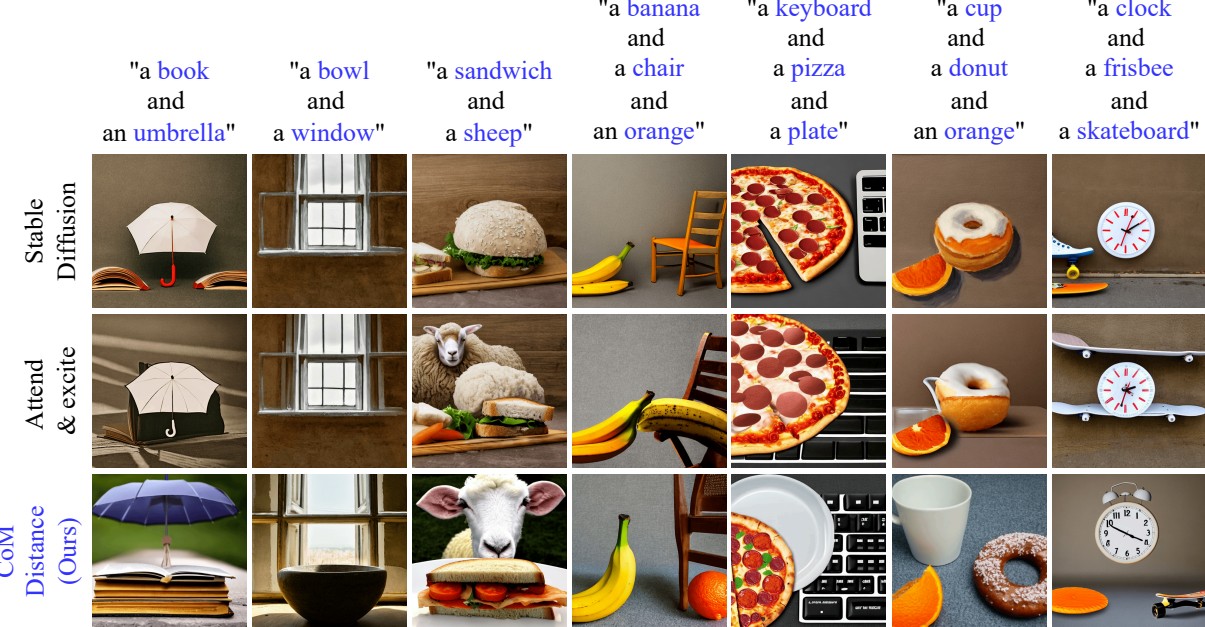

Figure 1: Comparison of compositional generation capabilities between Stable Diffusion, Attend-and-Excite, and one of our proposed overlap-based methods (CoM Distance) for textual prompts containing two and three entities while using SD-1.4 as the backbone: Stable Diffusion (first row) and Attend-and-Excite (second row) often fail to generate all the specified entities in the input prompt, a problem known as entity missing. Our training-free approach (third row) addresses this issue by employing an overlap-based objective function (CoM Distance) on cross-attention maps during the denoising steps, resulting in a more faithful generation of all the entities mentioned in the input prompt.

In recent years, text-to-image (T2I) diffusion-based (Ho et al., 2020) models such as Stable Diffusion (Rombach et al., 2022; Podell et al., 2023) and DALL-E (Ramesh et al., 2021; 2022; Betker et al., 2023) have shown promising performance in generating realistic, creative, diverse, and high-quality images from textual descriptions. These models leverage the iterative denoising process, along with text embeddings through the cross-attention mechanism, to generate images used in many real-world applications across various domains, such as controllable image synthesis (Ho & Salimans, 2021; Epstein et al., 2023; Zheng et al., 2023), image editing (Kawar et al., 2023; Zhang et al., 2023), 3D scene production (Poole et al., 2023; Lin et al., 2023; Richardson et al., 2023), video generation (Ho et al., 2022a;b).

However, despite their impressive capabilities, these models often struggle to faithfully capture all the entities, attributes, and relationships described in the input prompt, leading to various compositional misalignments such as *entity missing* (Chefer et al., 2023; Sueyoshi & Matsubara, 2024; Zhang et al., 2024; Liu et al., 2022; Kim et al., 2023), *incorrect attribute binding* (Feng et al., 2023; Li et al., 2023; Rassin et al., 2023; Wang et al., 2024), *incorrect spatial relationships* (Chatterjee et al., 2024; Gokhale et al., 2022; Chen et al., 2024), and *numeracy-related problems* (Binyamin et al., 2024; Zafar et al., 2024).

Entity missing, in particular, is a prevalent issue where the model fails to faithfully generate one or more entities specified in the textual prompt, significantly hurting the compositional generation performance and limiting the practical usability of T2I models. Previous works have explored both *training-based* (Ruiz et al., 2022; Wen et al., 2023; Kumari et al., 2022; Zarei et al., 2024) and *inference-based* Chefer et al. (2023); Sueyoshi & Matsubara (2024); Zhang et al. (2024); Liu et al. (2022); Kim et al. (2023); Karthik et al. (2023); Eyring et al. (2024) approaches to mitigate this problem. While fine-tuning the model on carefully curated datasets can improve compositional alignment, it is computationally expensive, may not generalize well to

unseen concepts, and can potentially introduce undesirable properties compared to the primal training phase. On the other hand, inference-time approaches that manipulate the intermediate latent code by optimizing an objective function over cross-attention maps between the input tokens and the image regions have shown promise in addressing entity missing without requiring model retraining. However, there has been a lack of systematic investigation into the optimization objectives that are most effective for this task.

In this work, we thoroughly examine three potential sources of the entity missing problem from the perspective of cross-attention maps between input tokens and image regions: (1) insufficient attention intensity, (2) excessive attention spread, and (3) significant overlap between attention maps of different entities. We discovered that while the two former phenomena can contribute to the entity missing problem, the latter is the root cause. More precisely, through extensive empirical analysis, we demonstrate that overlap-based metrics that quantify the pixel-level overlap between attention maps of different entities are highly predictive of entity missing. Building on this insight, we propose four overlap-based objective functions that can optimize the latent code of the diffusion model during inference: *Intersection over union (IoU)*, *center-of-mass (CoM) distance*, *Kullback–Leibler (KL) divergence*, and *clustering compactness (CC)*. By minimizing these objectives, our method effectively reduces the overlap between attention maps of entities and improves the model's ability to generate all the entities specified in the textual prompt faithfully.

We conducted a series of experiments using a synthetic set of prompts encompassing all combinations of object categories derived from the COCO dataset (Lin et al., 2014), which we refer to as COCO-Comp. For evaluation, we employed both human judgments and automatic metrics, including visual question answering (VQA) score (Hu et al., 2023; Li et al., 2022), captioning score (Li et al., 2022), and CLIP score Hessel et al. (2021). As shown in Figure 1, our results demonstrate that the proposed overlap-based optimization method significantly outperforms previous inference-time approaches, achieving state-of-the-art performance in compositional alignment. Notably, we found that optimizing the CoM distance between attention maps produced the best outcomes, increasing the success rate of entity generation by about 7% in scenarios with two entities and by more than 17% in scenarios with three entities, compared to existing state-of-the-art methods.

To validate the robustness of our overlap-based metrics, we tested our approach using a diverse set of model backbones containing SD-1.4, SD-2 (Rombach et al., 2022), SD-XL (Podell et al., 2023) and well-established compositional generation benchmarks, such as T2I-CompBench (Huang et al., 2023) and HRS-Bench (Bakr et al., 2023), in addition to the freestyle captions of COCO dataset, all demonstrating the superiority of our overlap-based approach over baselines. Furthermore, manipulating the latent code during the denoising process can lead to undesirable effects, such as reduced image quality and unnatural appearance. To address this issue, we assessed our methods using the Frechet Inception Distance (FID) score (Heusel et al., 2017), Coverage, and Density (Naeem et al., 2020), all demonstrating only a slight degradation in quality compared to the best-performing baseline. Finally, we evaluated the effectiveness of the proposed overlap-based methods in addressing other challenges in visual compositional generation, including incorrect attribute binding, incorrect spatial relationship, and numeracy-related issues. The results demonstrated that solely improving entity preservation can effectively contribute to mitigating these compositional generation challenges.

Hence, our primary contributions include a systematic investigation into three potential causes of entity missing in T2I diffusion models, identifying the overlap between attention maps of entities as the most critical factor. Building on the insights gained from this analysis, we introduce four novel overlap-based objective functions for inference-time optimization of latent codes during the denoising process. Lastly, comprehensive experiments across diverse model backbones and benchmarks demonstrate that our proposed methods significantly surpass previous approaches in various compositional alignment metrics.

## 2 Related Works

### 2.1 Visual Compositional Generation

Compositionality, the study of how basic components combine to form entirely new concepts (Peters et al., 2017), is a fundamental characteristic of the natural world and a crucial aspect of human cognition, allowing individuals to reason and adapt to novel situations (Phillips & Wilson, 2010; Frankland & Greene, 2020;

Reverberi et al., 2012). Building on this concept, compositional generation has emerged as a significant area of research in recent years, particularly in developing T2I models. Numerous studies have focused on advancing and evaluating T2I models to improve and assess their ability to handle compositional generation (Huang et al., 2023; Bakr et al., 2023; Li et al., 2024). These studies indicate that despite the impressive capabilities of T2I models in generating realistic and high-quality images, they sometimes fail to produce outputs that fully align with the provided textual input. This misalignment is particularly evident when the model encounters complex or novel compositions poorly represented in the training data, leading to compositional generation failures such as *entity missing*, *incorrect attribute binding*, *incorrect spatial relationships*, and *numeracy-related issues*.

Entity missing occurs when at least one entity specified in the textual prompt is not completely represented in the generated image (Chefer et al., 2023; Sueyoshi & Matsubara, 2024; Zhang et al., 2024; Liu et al., 2022; Kim et al., 2023). For example, considering the prompt "A cat sitting on a chair next to a dog", the generated image may only contain the chair, with the cat either missing entirely or only partially visible. This problem typically arises when the model does not allocate sufficient attention to certain entities during the generation process, leading to their partial or total absence in the final output (Chefer et al., 2023). On the other hand, incorrect attribute binding refers to the mis-association of an attribute, such as color, texture, shape, and size, with the corresponding entity (Feng et al., 2023; Li et al., 2023; Rassin et al., 2023; Wang et al., 2024; Wu et al., 2023; Park et al., 2024). For example, in response to the prompt "A green apple and a red ball" the T2I model might generate an image where the apple is red and the ball is green, incorrectly swapping the attributes. This problem often stems from improper alignment in the attention maps of the entity and its corresponding attribute during the generation process.

Moreover, incorrect spatial relationships occur when the relative positioning of entities in the generated image does not correspond to the spatial arrangement specified in the input text (Chatterjee et al., 2024; Gokhale et al., 2022; Chen et al., 2024; Feng et al., 2024; Yang et al., 2022). For example, if the prompt states, "A person standing behind a car," the model might generate an image with the person in front of the car instead. This problem often arises from the model's limited capacity to interpret and apply geometric relationships such as depth, perspective, or relative positioning. Finally, numeracy-related problems occur when the number of instances of a specific entity in the generated image does not match the count specified in the prompt (Binyamin et al., 2024; Zafar et al., 2024; Kang et al., 2023). For example, if the prompt is "Three dogs playing in a park," the model might generate an image with only two dogs or more than three, failing to meet the specified number. This issue often results from the model's poor ability to reason about numbers and generalize beyond the examples it encountered during the training phase.

## 2.2 Training-Free Methods for Compositional Generation

While several fine-tuning-based approaches (Sun et al., 2023; Jiang et al., 2024; Zarei et al., 2024; Guo et al., 2024; Wen et al., 2023) have been proposed to mitigate failure modes in visual compositional generation, these methods are often resource-intensive, requiring the careful curation of appropriate datasets. Additionally, they may introduce new challenges, such as overfitting to the fine-tuning data and the potential loss of beneficial properties inherent in the T2I model from its initial training phase.

On the other hand, training-free methods provide a promising, on-the-fly solution for addressing compositional generation failures, such as missing entities, in T2I diffusion models. These approaches eliminate the need for costly fine-tuning while avoiding its potential drawbacks while preserving the generation capability of the original model. Some of these methods leverage the *multiple generation* paradigm (Karthik et al., 2023; Liu et al., 2024), allowing the T2I model to produce several parallel latent codes during the denoising process. The optimal result is then selected based on an evaluation metric applied either to the final image or to intermediate latent embeddings. However, generating multiple instances significantly increases inference time and makes these methods resource-intensive.

A separate line of research focuses on enhancing compositional alignment between the input text and the generated image by manipulating intermediate latent embeddings during the denoising process. This approach, known as *generative semantic nursing (GSN)* (Chefer et al., 2023), involves defining a loss function over the cross-attention maps between input tokens and image regions, as these maps encode valuable spatial

information about entities and attributes. By updating the latent embeddings in the opposite direction of the loss function's gradient, compositional generation is expected to improve.

For example, the *Attend-and-Excite* method (Chefer et al., 2023) employs an intensity-based loss function designed to prevent the weakening of the entity with the lowest maximum attention score, while *Divide-and-Bind* (Li et al., 2023) uses a Kullback–Leibler (KL) divergence loss function to ensure the proper association of attributes with corresponding entities. Additionally, *Predicated Diffusion* (Sueyoshi & Matsubara, 2024) interprets the intended meaning of the input prompt as propositions based on predicate logic, leading to a differentiable loss function that guides the image generation process to better satisfy the propositions. Lastly, *Attention Regulation* (Zhang et al., 2024) formulates a constrained optimization problem to minimize attention mismatches, aligning attention maps more closely with the input text prompt.

On the other hand, training-free approaches such as *Structured Diffusion* Feng et al. (2023) incorporate linguistic structures obtained from constituency trees directly into the cross-attention layers. More precisely, they extract noun phrases from the input text prompt, feed them separately to the CLIP text-encoder to obtain corresponding embedding, and augment the value matrix of the cross-attention mechanism with the obtained embeddings to promote the role of noun phrases.

Notably, a number of studies have taken the complexity of input prompts into account by adjusting inference time according to the intricacy of the input. For example, *Composable Diffusion* (Liu et al., 2022) introduces a method that first divides the input text prompt into distinct concepts, using the "and" token as a delimiter. During each denoising step, multiple instances of the denoiser independently generate latent embeddings corresponding to each concept. These latent embeddings are then combined to form a unified latent code, ultimately producing an image that integrates all specified concepts.

## 3 Preliminaries

### 3.1 Latent Diffusion Model

Latent diffusion models (LDMs) (Rombach et al., 2022) are a class of generative models designed to map a latent representation of an image to the corresponding high-dimensional pixel space. In contrast to traditional diffusion models that operate directly in pixel space, LDMs first encode an image $x$ into a lower-dimensional latent space using an encoder $E$, and then apply the diffusion process on the latent embedding $z = E(x)$. To reconstruct the image from the denoised latent representation, a decoder $D$ maps the denoised latent back to pixel space, generating the final image $\hat{x}$.

The diffusion process in LDMs consists of a forward (diffusion) process and a reverse (denoising) process. In the forward process, a Gaussian noise ($\epsilon$) is gradually added to the latent embedding ($z$) over a fixed number of time steps, transforming the original latent $z_0$ into a noisy latent $z_t$ at timestep $t$. This forward process can be described as equation 1.

$$z_t = \sqrt{\bar{\alpha}_t} z_0 + \sqrt{1 - \bar{\alpha}_t}\epsilon, \quad \epsilon \sim \mathcal{N}(0, I) \tag{1}$$

where $\bar{\alpha}_t = \prod_{i=1}^{t} \alpha_i$, and $\alpha_i$ are predefined noise schedule parameters that control the amount of noise added at each time step.

In the reverse process, the noisy latent $z_t$ is gradually denoised using a learnable denoising network $\epsilon_\theta$ parameterized by a U-Net architecture (Ronneberger et al., 2015). More precisely, the denoising process starts from a latent representation sampled from a standard Gaussian distribution, i.e., $z_T \sim \mathcal{N}(0, I)$, and iteratively refines it using equation 2.

$$z_{t-1} = \frac{1}{\sqrt{\alpha_t}}\left(z_t - \frac{1 - \alpha_t}{\sqrt{1 - \bar{\alpha}_t}}\epsilon_\theta(z_t, t)\right) + \sigma_t\epsilon, \quad \epsilon \sim \mathcal{N}(0, I) \tag{2}$$

where $\sigma_t$ is the standard deviation of the Gaussian noise added at timestep $t$, and $\epsilon_\theta(z_t, t)$ is the output of the denoiser netwrok.

The denoiser is trained using a simple loss function that minimizes the $L_2$ distance between the predicted noise $\epsilon_\theta$ and the ground truth noise $\epsilon$ as described in equation 3.

$$\mathcal{L} = \mathbb{E}_{x_0,t,\epsilon} \left[ \| \epsilon - \epsilon_\theta(\sqrt{\bar{\alpha}_t} E(x_0) + \sqrt{1 - \bar{\alpha}_t}\epsilon, t) \|^2 \right] \tag{3}$$

where $x_0$ is the original image, $t$ is a timestep sampled uniformly from $\{1, \ldots, T\}$, and $\epsilon \sim \mathcal{N}(0, I)$.

The use of a lower-dimensional latent space in LDMs offers several advantages over pixel-space diffusion models. First, it allows for more efficient training and inference, as the diffusion process operates on a more compact representation. Second, the latent space provides a more semantically meaningful representation of the image, which can lead to improved sample quality and controllability (Rombach et al., 2022).

## 3.2 Text-to-Image Diffusion Models

T2I diffusion models such as Stable Diffusion (Rombach et al., 2022) extend the capabilities of latent diffusion models by conditioning the generation process on textual descriptions. This allows for creating images that accurately reflect the semantic content provided in the input text. The conditioning is typically achieved through the use of cross-attention layers within the denoising U-Net architecture.

More precisely, given an input text prompt, the first step in a T2I diffusion model is to obtain the text embedding $P \in \mathbb{R}^{N \times D_P}$ using a pre-trained text encoder, such as CLIP (Radford et al., 2021), where $N$ is the maximum context length and $D_P$ is the dimensionality of the text embedding space. Then, at each denoising step $t$, the cross-attention mechanism takes the text embedding $P$ as the key and value matrices and the noisy latent representation $z_t$ as the query matrix to produce cross-attention maps according to equation 4.

$$A_t = \text{Softmax}\left( \frac{W_Q z_t P^T W_K^T}{\sqrt{d_k}} \right) W_V P \tag{4}$$

where $A_t \in \mathbb{R}^{H \times W \times N}$ is the resulting cross-attention map at time step $t$. Moreover, $W_Q$, $W_K$, and $W_V$ are learnable weight matrices for the query, key, and value, respectively, and $d_k$ is a scaling factor typically set to the square root of the key dimensionality.

Notably, $A_t[i, j, n]$ represents the attention score between the latent representation at spatial location $(i, j)$ and the $n$-th text token. These attention scores indicate the relevance of each text token to different regions of the image being generated. Through this work, we denote the attention map corresponding to the entity $e$ at time step $t$ as $A_t^e \in \mathbb{R}^{H \times W}$.

## 4 When and Why Does Entity Missing Happen?

The entity missing problem, where one or more entities mentioned in the input text prompt are not faithfully generated in the output image, is a significant challenge in T2I diffusion models. Understanding the potential causes of this problem is crucial to developing effective solutions to compositional generation issues. In this section, we analyze the entity missing problem from the perspective of cross-attention maps, which capture the relationships between image regions and text tokens.

We hypothesize that the entity missing problem may stem from three key factors related to cross-attention maps: (1) *attention intensity*, (2) *attention spread*, and (3) *attention overlap*. The first two factors focus on the evolution of the attention scores of each entity separately during denoising steps, while the third factor investigates how two or more entities affect and interact with each other. To examine these factors, we introduce several metrics that quantify different aspects of cross-attention maps, such as attention intensity ($Int$), attention spread ($Var$), intersection over union ($IoU$), center-of-mass (CoM) distance ($D_{CoM}$), Kullback–Leibler (KL) divergence ($D_{KL}$), and clustering compactness ($CC$). We then conduct several experiments on a diverse set of prompts and analyze the correlation of each metric with the success rate of the T2I model to generate entities mentioned in the prompts.

### 4.1 Attention Intensity of an Entity

Attention intensity refers to the average strength of the attention scores assigned to a specific entity token across different spatial locations in the cross-attention map. We hypothesize that insufficient attention intensity could lead to the entity missing problem, as the model may fail to allocate enough attention to the corresponding entity during the image generation process. To quantify attention intensity, we propose the *Int* metric, which measures the average attention score of an entity $e$ across all spatial locations at a given time step $t$, according to equation 5.

$$Int(A_t^e) = \frac{1}{H \times W} \sum_{i,j} A_t^e[i,j] \tag{5}$$

where $A_t^e$ is the attention map of entity $e$ at time step $t$, and $H$ and $W$ are the height and width of the attention map, respectively.

Figure 2 illustrates how attention intensity evolves over time for the entity "bird" in two different scenarios. In the successful case, the attention intensity of the entity "bird" starts from a higher value and remains relatively high throughout the generation process compared with the failure case, enabling the model to faithfully generate the entity "bird" in the output image. This insightful example highlights the role of attention intensity, especially in the initial steps of the denoising process, in avoiding entity missing problems.

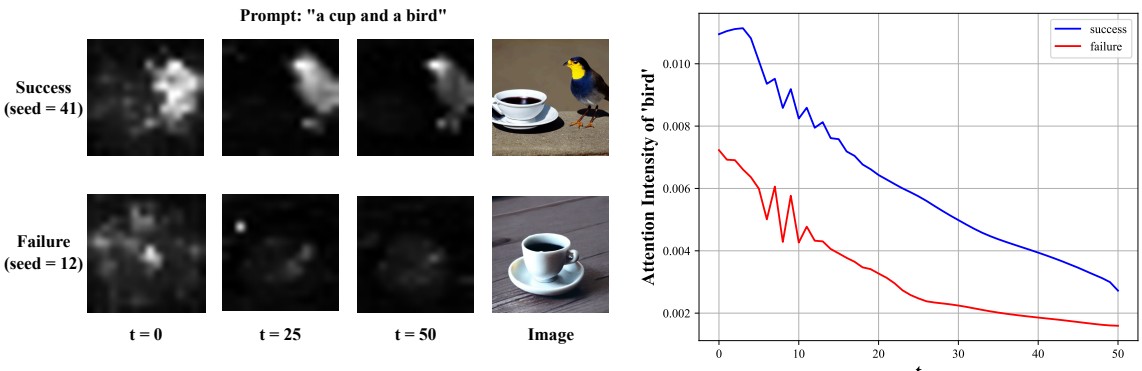

Figure 2: **Left:** Three attention maps of entity "bird" are drawn at different time steps (0, 25, and 50) along with the generated image for both success and failure cases. **Right:** While in both success and failure cases, the attention intensity (Equation 5) of entity "bird" decreased over time, the bad initialization of $z_T$ with $seed = 12$ resulted in the vanishing of the attention scores of "bird" at $t = 50$.

### 4.2 Attention Spread of An Entity

In addition to attention intensity, the spread of attention scores related to an entity across different spatial locations in the cross-attention map can also contribute to the entity missing problem. We hypothesize that excessive attention spread could lead to the entity missing problem, as the model may struggle to generate a coherent representation of the entity in the output image. To quantify the attention spread of an entity, we introduce the $Var$ metric, which measures the variance of attention scores around the center of mass ($CoM$) of its cross-attention map as described in equation 6.

$$Var(P_t^e) = \sum_{i,j} P_t^e[i,j] \times \left\lVert CoM(P_t^e) - \begin{bmatrix} i \\ j \end{bmatrix} \right\rVert_2^2 \tag{6}$$

where $P_t^e$ is the normalized attention map of entity $e$ at time $t$ in such a way that for each spatial location $(i, j)$, $P_t^e[i,j] = \frac{A_t^e[i,j]}{\sum_{k,h} A_t^e[k,h]}$, and $CoM(P_t^e) = \sum_{i,j} P_t^e[i,j] \cdot \begin{bmatrix} i \\ j \end{bmatrix}$ represents the center of mass of the normalized attention map, calculated as the weighted average of spatial coordinates using the normalized attention scores as weights.

As illustrated in Figure 3, the failure case exhibits significantly higher attention spread in terms of the $Var$ metric compared to the success case, meaning the attention scores are dispersed across a wider area around the $CoM$. We believe that the excessive spread hinders the model's ability to coherently represent the entity "snowboard", leading to the entity missing problem.

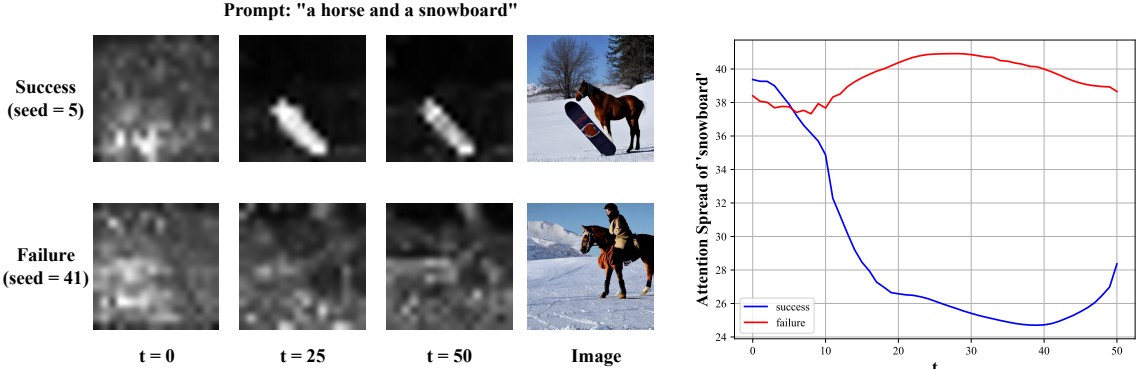

Figure 3: **Left:** Three attention maps of entity "snowboard" are drawn at different time steps (0, 25, and 50) along with the generated image for both success and failure cases. **Right:** During most of the time steps, the attention spread (Equation 6) of entity "snowboard" in the failure case is much higher than in the success case, resulting in the entity missing problem.

### 4.3    Attention Overlap Between Entities

From the viewpoint of the cross-attention mechanism, each entity token of the input prompt attends to spatial locations during denoising steps to form the corresponding entity in the final image. This process induces an implicit competition between the entities to assign spatial locations to themselves. However, because of the unavoidable randomness in the latent initialization and the drawbacks of the CLIP text-encoder in representing complex prompts, some entities may not be able to properly attend to enough adjacent spatial locations to form the entity through the competition, resulting in the entity missing problem.

As illustrated in Figure 4, in the successful case, the attention overlap between the entities "cat" and "vase" remains relatively low throughout the denoising process, which allows the T2I model to generate a coherent representation of both entities in the output image. In contrast, the failure case exhibits a significantly higher attention overlap, with the attention maps of the two entities competing for the same spatial regions. This competition leads to one entity (in this case, the "cat") being underrepresented or missing in the generated image.

One way to study this competition is by measuring how much the attention maps of different entities overlap with each other across spatial locations. Hence, we propose four overlap-based metrics: *intersection over union (IoU)*, *CoM distance*, *symmetric KL divergence*, and *clustering compactness (CC)*, as described in table 1.

The **IoU** metric explicitly measures the degree of overlap between the attention maps of two entities by calculating the ratio of the intersection of their attention scores to the union of their attention scores across all spatial locations, where the intersection and union are computed using the product and summation operations, respectively.

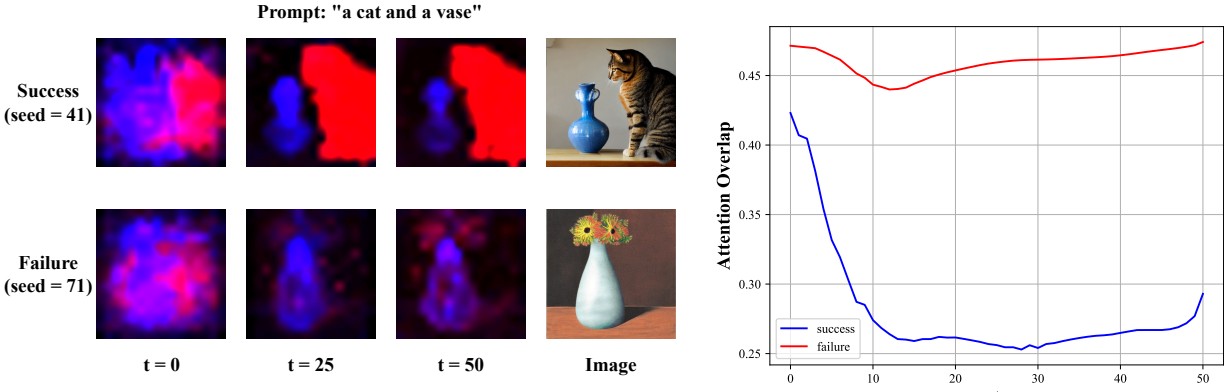

Figure 4: **Left:** For both the success and the failure cases and at three time steps (0, 25, and 50), the attention maps of entities 'cat' and 'vase' are depicted together in one image, with the attention scores of cat and vase being red and blue, respectively. **Right:** During most of the time steps, the attention overlap measured by $IoU$ metric (Table 1) between two entities in the failure case is much higher than that in the success case, resulting in the missing entity "cat" in the generated image.

Table 1: Four overlap-based metrics are formally introduced, where $P_t^{e_1}$ and $P_t^{e_2}$ are the normalized version of attention maps $A_t^{e_1}$ and $A_t^{e_2}$ corresponding to two entities $e_1$ and $e_2$. For the sake of simplicity, we defined the metrics over only two entities. However, the extended version of these metrics is available in the Appendix.

| Metric | Formulation |
|---|---|
| Intersection over Union (IoU) | $IoU(P_t^{e_1}, P_t^{e_2}) = \sum_{i,j} \frac{P_t^{e_1}[i,j] \times P_t^{e_2}[i,j]}{P_t^{e_1}[i,j] + P_t^{e_2}[i,j]}$ |
| CoM Distance | $D_{CoM}(P_t^{e_1}, P_t^{e_2}) = \|CoM(P_t^{e_1}) - CoM(P_t^{e_2})\|_2^2$ |
| Symmetric KL Divergence | $D_{KL}(P_t^{e_1}, P_t^{e_2}) = \frac{1}{2} \sum_{i,j} P_t^{e_1}[i,j] \log \frac{P_t^{e_1}[i,j]}{P_t^{e_2}[i,j]} + P_t^{e_2}[i,j] \log \frac{P_t^{e_2}[i,j]}{P_t^{e_1}[i,j]}$ |
| Clustering Compactness (CC) | $CC(P_t^{e_1}, P_t^{e_2}) = \frac{1}{H \times W} \sum_{i,j} \max\{P_t^{e_1}[i,j], P_t^{e_2}[i,j]\}$ |

On the other hand, the **CoM distance** measures the divergence in the focal points of attention maps using Euclidean distance. More precisely, it quantifies the spatial separation between the CoM of the attention maps, indicating how far apart the entities are focused on average.

Moreover, inspired by KL divergence in information theory, we propose the **symmetric KL divergence** between two normalized attention maps, where each normalized attention map is considered a probability distribution of attention scores over spatial locations. More precisely, this metric captures how similarly the entities allocate attention across the spatial locations from a probabilistic perspective.

Finally, **clustering compactness (CC)** is a measure usually used in unsupervised machine learning to assess the tightness of clusters formed by grouping data points based on their similarity in a latent space. We believe that the cross-attention mechanism during denoising steps is similar to a clustering process, where each key vector related to an entity is a center, and each query vector associated with the latent embedding at a spatial location is a data point in the clustering process. Considering this analogy, we propose the CC metric on the attention maps of two entities as the intra-cluster similarity where the similarity is measured by the inner-product operation (Table 1). Note that the inner product of the cross-attention mechanism between the key vector related to the entity $e$ and the query vector associated with the spatial location $(i, j)$ is $A_e^t[i,j]$. Moreover, for each query vector (data point), the $max$ operation in the CC metric selects the most

similar (nearest) key vector (centers), similar to a clustering process. Put them all together, the $CC$ metric quantifies the degree to which spatial locations are fully assigned to a single entity, promoting distinct and well-separated entity representations.

### 4.4 Attention Overlap Is the Root Cause of Entity Missing

We analyzed the relationship between the model's success rate in faithfully generating entities and each introduced metric by running the Stable Diffusion version v1.4 model on validation set of our proposed dataset COCO-Comp, which contains 190 prompts including 20 entities in the format of "a {entity} and a {entity}", where entities are from the COCO dataset and the success rate is measured using a visual question-answering (VQA) approach called TIFA-score Hu et al. (2023). More details about the COCO-Comp dataset are provided in the Appendix A. Notably, when dealing with prompts containing more than one entity, the $Int$ metric is the minimum of $Int$ over entities, and the $Var$ metric is the average of $Var$ across all entities.

As illustrated in Figures 5, an unignorable correlation exists between the performance of the T2I model in avoiding entity missing and the proposed metrics, especially the overlap-based ones. We believe that the lower overlap value between two attention maps measured by $IoU$ and $D_{KL}$ induces discrepancy in the attention distribution of entities and leads the entities to focus on distinct regions rather than being conflated, resulting in a higher VQA score. Moreover, a higher distance between the $D_{CoM}$ of entities implicitly makes more room for each entity to attend to different regions of the image without worrying about overshadowing or masking others. Similarly, higher $CC$ values imply that at each spatial location, $(i, j)$, one entity completely dominates others in the sense of attention scores, which results in less possibility of the entity missing.

Moreover, while Figure 5 demonstrates that higher attention intensity ($Int$) and lower attention spread ($Var$) results in a reduced possibility of entity missing in the generated image, a weaker correlation value between these metrics and the success rate of compositional generation is reported compared to the overlap-based metrics. These observations underscore the importance of minimizing overlap between cross-attention maps of entities compared to other metrics, as it enables the weaker entity—characterized by lower attention scores across overlapped pixels—to evade dominance by the stronger one.

## 5 Method

The investigation presented in Section 4.4 aimed to clarify the relationship between various metrics applied to cross-attention maps and the occurrence of missing entities. The analysis revealed compelling evidence highlighting the importance of overlap-based metrics, such as IoU, $D_{CoM}$, $D_{KL}$, and CC, in reliably assessing performance related to mitigating missing entities. Building on insights from prior studies (Chefer et al., 2023; Li et al., 2023; Sueyoshi & Matsubara, 2024; Zhang et al., 2024) and leveraging the advantages of training-free methods, we propose a loss function-based approach. This method minimizes a loss function during denoising steps to reduce the overlap between the attention maps of different entities, thereby improving the model's ability to faithfully represent all entities mentioned in the input text prompt.

Specifically, for a prompt containing two entities $e_1$ and $e_2$, an overlap-based loss function $\mathcal{L}$ is computed over the normalized attention maps of these entities ($P_t^{e_1}$ and $P_t^{e_2}$) at each denoising step $t$. The loss function $\mathcal{L}(P_t^{e_1}, P_t^{e_2})$ can take various forms, including $IoU(P_t^{e_1}, P_t^{e_2})$, $-D_{CoM}(P_t^{e_1}, P_t^{e_2})$, $-D_{KL}(P_t^{e_1}, P_t^{e_2})$, or $-CC(P_t^{e_1}, P_t^{e_2})$.

Following the generative semantic nursing paradigm introduced in Chefer et al. (2023), in order to minimize the loss function $\mathcal{L}$, we update the latent representation $z_t$ in the direction opposite to the gradient of $\mathcal{L}$ with respect to $z_t$ at each denoising step $t$. This iterative update process is performed using the gradient descent algorithm, as described in equation 7.

$$z_t \leftarrow z_t - \alpha \nabla_{z_t} \mathcal{L} \tag{7}$$

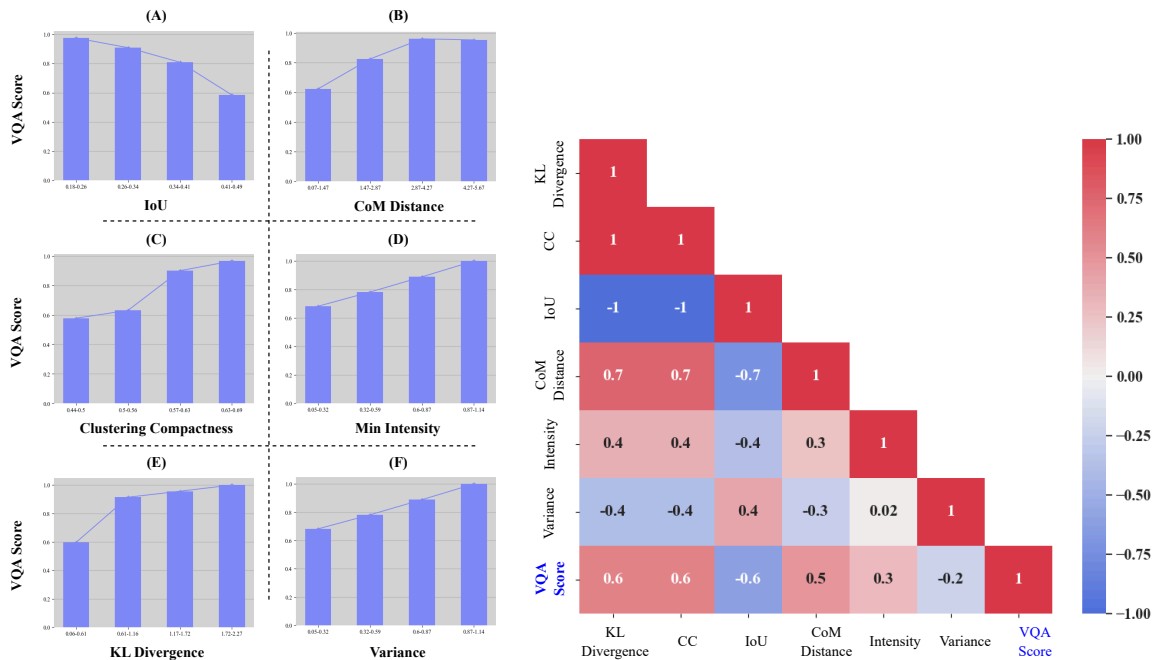

Figure 5: **Left:** The relationships between the proposed metrics and the VQA score: A higher value of $CoM$ distance, $KL$ divergence, and $CC$, and attention intensity results in a higher VQA score, while a lower value of $IoU$ and attention spread ($Var$) improves the possibility of faithfully generating entities. **Right:** The correlation matrix showing the correlation values between the proposed metrics and the VQA score: The results reveal strong correlations between the overlap-based metrics ($IoU$, $CoM$ distance, $KL$ divergence, and $CC$) and the success rate of the model in faithfully generating the entities. In particular, the $CoM$ distance, $KL$ divergence, and $CC$ metrics exhibit strong positive correlations, while the $IoU$ metric shows a strong negative correlation with the VQA score. Moreover, there is a strong correlation among each pair of the overlap-based metrics as well.

where $\alpha$ denotes the learning rate, and $\nabla_{z_t}\mathcal{L}$ represents the gradient of the loss function $\mathcal{L}$ with respect to the latent embedding $z_t$.

Notably, these overlap-based metrics were selected for their diverse perspectives on the attention overlap issue, thereby providing a comprehensive understanding of the problem. For instance, $IoU$ and $D_{CoM}$ are computed in spatial space, where $IoU$ emphasizes the reduction of common areas between entities, while $D_{CoM}$ focuses on increasing the distance between them. On the other hand, $D_{KL}$ is a probabilistic measure that interprets the attention map as a probability density function, quantifying the distance between distributions. Lastly, $CC$ is a novel metric that operates in the feature space, derived from the concept of clustering compactness.

Minimizing the overlap between the attention maps of entities has at least two distinct yet interconnected benefits, which can help mitigate the entity missing problem. Firstly, by reducing the attention overlap, the weaker entity, which is recognized by lower attention scores across overlapped spatial locations, can avoid being dominated by the stronger entity, thus lessening the probability of one entity vanishing. Secondly, the reduction in overlap engenders a broader spatial coverage across the image for all entities. With reduced overlap between attention maps, entities have greater freedom to focus on adjacent pixels from various regions of the image, minimizing concerns about interference with other entities. The evolution of attention maps during denoising steps for each of the introduced overlap-based losses can be found in Appendix F.

Note that while the described metrics and loss functions can be easily extended to more than two attention maps, we have presented the case where the functions are applied to two attention maps only for the sake of simplicity. More information about the extension of these overlap-based metrics is available in the Appendix C.2.

# 6 Results

## 6.1 Experimental Setup

**Datasets.** We evaluate our approach on four datasets: (1) our set of carefully curated prompts called COCO-Comp featuring 2 to 4 entities in a structured format, (2) a subset of 1000 captions from the COCO dataset (Lin et al., 2014) with more complex and varied linguistic structures, (3) the validation section of T2I-CompBench (Huang et al., 2023), and (4) the validation section of HRS-Bench (Bakr et al., 2023). For COCO-Comp, we generated prompts using 71 entities obtained from the COCO validation set, strictly split into validation and test sets so that there are no overlapping entities between them, which is neglected in many previous works. This resulted in 190 validation prompts used for analysis (Section 4) and 1275 test prompts used for main experiments in the two-entity case. Moreover, we randomly selected 1000 test prompts for three-entity cases. More details on the prompt generation process and the complete list of entities in the COCO-Comp dataset can be found in Appendix A.

**Evaluation Metrics For Compositional Alignment.** We employed a diverse set of evaluation metrics to comprehensively assess the compositional alignment of generated images with the input prompts (Appendix B). These include:

1. Soft and harsh **human evaluation**, where in soft case raters assess each entity in the prompt as fully present (1.0), partially present (0.5), or missing (0.0) from the generated image, while in harsh case, the partially present state is considered as missing;

2. Soft and harsh **visual question answering (VQA)**, which are computed using the BLIP Li et al. (2022) and TIFA Hu et al. (2023) models. In the harsh evaluation, a score of 1.0 is awarded only if the VQA model accurately identifies all entities; otherwise, a score of 0 is assigned. On the other hand, in the soft VQA evaluation, the score is proportional to the number of entities that are generated in their entirety.

3. **CLIP similarity** (Hessel et al., 2021), which considers the cosine similarity between the prompt embedding and generated image embedding in the CLIP embedding space;

4. **Captioning score**, which measures the similarity between the prompt and the caption generated by the BLIP captioning model (Li et al., 2022) from the final image in the CLIP text embedding space (Hessel et al., 2021).

**Evaluation Metrics For Quality and Naturalness.**  Additionally, manipulating intermediate latent embeddings during the denoising process can cause the embeddings to fall outside the distribution expected by the U-Net denoiser, resulting in additional challenges such as unnatural artifacts and reduced image quality. To address this issue, we utilized the **Fréchet inception distance (FID) score** (Heusel et al., 2017) to evaluate the quality of images generated by T2I methods in comparison to real images from the COCO dataset (Lin et al., 2014). More precisely, the FID score measures the distance between the generated images by a T2I model and the real images in the feature space of a pre-trained Inception v3 network (Szegedy et al., 2016) in such a way that lower FID scores indicate higher quality and naturalness of the generated images. Furthermore, we adopted the **Coverage** and **Density** (Naeem et al., 2020) evaluation metrics which measure the diversity and quality of the generated images compared to the COCO real images (Lin et al., 2014) as reference, respectively. More precisely, A higher value of Density indicates better fidelity and quality, showing that the generated images are concentrated in regions where reference images (COCO images) are densely packed. A higher value of Coverage indicates better diversity, showing that the generated images capture the variability of the reference images (COCO images).

**Hyperparameters.**  We initially set the learning rate to 20 for our overlap-based optimization approach and decreased it linearly. Based on the validation set performance, the number of optimization steps per denoising step is set to 1. Moreover, we applied the gradient descent-based optimization at the first 25 denoising steps, while models are run for 50 denoising steps with guidance scale 7.5 and $512 \times 512$ image resolution for Stable Diffusion v1.4 and v2, and $1024 \times 1024$ image resolution for Stable Diffusion XL. The latent noise schedule is set to the default configuration in the Stable Diffusion codebase.

**Baselines.**  We compared our method to several existing training-free methods, including state-of-the-art ones for improving compositional generation in T2I diffusion models, including *Attend-and-Excite* (Chefer et al., 2023), *Divide-and-Bind* (Li et al., 2023), *Structured Diffusion* (Feng et al., 2023), *Predicated Diffusion* (Sueyoshi & Matsubara, 2024), *Attention-Reg* (Zhang et al., 2024), and *Composable Diffusion* Liu et al. (2022). Moreover, we evaluated our method against a well-known fine-tuning-based approach, *Dense Diffusion* (Kim et al., 2023), which incorporates user-defined layouts along with the textual input. It is important to note that our proposed overlap-based approach requires no additional input and functions as a plug-in module for any diffusion-based backbone. For all baseline methods, we utilized the official implementations provided by the authors, following their recommended hyperparameter settings. Further details about each baseline are provided in Section 2.2 and Appendix C.1.

## 6.2  Quantitative Results

**COCO-Comp Dataset.**  Tables 2 and 3 present the main quantitative results of our proposed overlap-based methods on the Stable Diffusion v1.4 model compared to several state-of-the-art baselines on the test set of two-entity and three-entity prompts from the COCO-Comp dataset, respectively. Our method significantly outperforms the vanilla Stable Diffusion model and all baselines on every metric, demonstrating the effectiveness of overlap-based optimization for mitigating entity missing issues. Particularly in the two-entity case, using the CoM distance as the overlap metric yields the best results, with an absolute improvement of **23.3%** in soft human score and **24.1%** in soft TIFA score over the Stable Diffusion baseline. Notably, the loss function $D_{CoM}$ surpasses the strongest baseline, Predicated Diffusion, by **9%** in soft human score and **3.6%** in soft TIFA score. The other overlap-based metrics, $IoU$, $CC$, and $D_{KL}$, also perform comparably to $D_{CoM}$, all outperforming other state-of-the-art methods. Importantly, our approach's effectiveness increases with prompt complexity. Moving from two to three-entity prompts in the COCO-Comp dataset, our overlap-based methods' performance advantage over the strongest baseline widens, with the soft TIFA score gap growing by **17.6%**, underscoring their ability to handle complex prompts.

Table 2: Quantitative results of our overlap-based methods and the state-of-the-art baselines on the Stable Diffusion v1.4 model as the backbone using the **two-entity** prompts from the COCO-Comp dataset. The CoM Distance method (underlined) significantly outperforms all baselines (best in bold), demonstrating the effectiveness of overlap-based latent optimization for mitigating the entity missing problem. Note that the letters 'S' and 'H' in both Human and TIFA evaluations denote soft and harsh, respectively.

| Method | Human Score (S) (↑) | Human Score (H) (↑) | TIFA Score (S) (↑) | TIFA Score (H) (↑) | Captioning Score (↑) | CLIP Similarity (↑) |
|---|---|---|---|---|---|---|
| Stable Diffusion | 57.5% | 54.7% | 69.9% | 40.0% | 0.789 | 0.297 |
| Attend-and-Excite | 69.6% | 57.9% | 87.6% | 76.5% | 0.811 | 0.312 |
| Divide-and-Bind | 68.7% | 56.5% | 86.5% | 73.6% | 0.815 | 0.309 |
| Structure Diffusion | 61.5% | 52.2% | 74.3% | 49.6% | 0.792 | 0.299 |
| Predicated Diffusion | **71.8%** | **60.3%** | **90.6%** | **81.6%** | **0.823** | 0.314 |
| Attention Regulation | 69.4% | 58.3% | 88.6% | 78.4% | 0.818 | **0.316** |
| Composable Diffusion | 47.9% | 36.8% | 66.2% | 37.6% | 0.761 | 0.297 |
| IoU | 78.5% | 67.6% | 92.7% | 85.3% | 0.824 | 0.317 |
| CoM Distance | **80.8%** | **70.2%** | **94.0%** | **88.2%** | **0.825** | **0.318** |
| KL Divergence | 77.9% | 66.3% | 91.7% | 83.5% | 0.818 | 0.316 |
| CC | 77.8% | 66.4% | 92.2% | 84.6% | 0.821 | 0.315 |

Table 3: Quantitative results of our overlap-based methods and the state-of-the-art baselines on the Stable Diffusion v1.4 model as the backbone using the **three-entity** prompts from the COCO-Comp dataset. The CoM Distance method (underlined) significantly outperforms all baselines (best in bold), demonstrating the effectiveness of overlap-based latent optimization for mitigating the entity missing problem. Note that the letters 'S' and 'H' in both TIFA and BLIP evaluations denote soft and harsh, respectively.

| Method | TIFA Score (S) (↑) | TIFA Score (H) (↑) | BLIP-VQA Score (S) (↑) | BLIP-VQA Score (H) (↑) | Captioning Score (↑) | CLIP Similarity (↑) |
|---|---|---|---|---|---|---|
| Stable Diffusion | 62.6% | 15.2% | 65.6% | 21.6% | 0.763 | 0.305 |
| Attend-and-Excite | 80.7% | 52.9% | **88.3%** | **69.1%** | 0.780 | 0.321 |
| Divide-and-Bind | **82.6%** | **56.0%** | 88.1% | 68.8% | 0.790 | 0.323 |
| Structure Diffusion | 63.1% | 16.6% | 67.6% | 23.5% | 0.757 | 0.306 |
| Predicated Diffusion | 82.6% | 53.1% | 87.2% | 64.2% | **0.800** | **0.327** |
| Composable Diffusion | 46.2% | 5.7% | 60.4% | 18.1% | 0.710 | 0.288 |
| IoU | 88.3% | 67.4% | 92.8% | 79.3% | **0.799** | **0.331** |
| CoM Distance | **89.9%** | **73.6%** | 93.3% | **83.3%** | 0.790 | 0.329 |
| KL Divergence | 86.7% | 63.4% | 91.1% | 75.2% | **0.799** | 0.330 |
| CC | 87.6% | 65.6% | **93.4%** | 81.2% | 0.792 | 0.328 |

**HRS-Bench and T2I-CompBench Datasets.** Additionally, a quantitative comparison between our overlap-based methods and the state-of-the-art training-free approach, Attend-and-Excite (Chefer et al., 2023), is presented in Tables 5 and 4, using the color-related prompts of well-known and challenging compositional generation benchmarks HRS-Bench (Bakr et al., 2023) and T2I-CompBench (Huang et al., 2023). The results clearly highlight the superiority of the proposed overlap-based methods across all evaluation metrics. Notably, the methods improved the harsh TIFA score by 28.6% and 3.3% over the HRS-Bench and the T2I-CompBench, respectively. Notably, both our proposed dataset, COCO-Comp, and the HRS-Bench include a mix of common (e.g., "cat and dog") and rare (e.g., "giraffe and pizza") entity combinations, whereas the T2I-CompBench primarily focuses on more common pairings. This highlights that our proposed method not only achieves improvements in natural compositions but also demonstrates greater enhancement in rare compositions, which are unlikely to be encountered during the training phase.

Table 4: Quantitative comparison between our overlap-based methods and the state-of-the-art training-free method, Attend-and-Excite, on the HRS-Bench where Stable Diffusion v2 is used as the backbone. The overlap-based approaches outperformed the baseline methods across all evaluation metrics.

| Method | TIFA Score (S) (↑) | TIFA Score (H) (↑) | BLIP-VQA Score (S) (↑) | BLIP-VQA Score (H) (↑) | Captioning Score (↑) | CLIP Similarity (↑) |
|---|---|---|---|---|---|---|
| Stable Diffusion | 58.7% | 23.2% | 63.8% | 28.7% | **0.660** | 0.332 |
| Attend-and-Excite | **64.5%** | **31.9%** | **70.8%** | **39.5%** | 0.645 | **0.335** |
| IoU | 81.1% | 52.3% | 83.7% | 57.3% | **0.688** | 0.345 |
| CoM Distance | **82.0%** | **60.5%** | **86.7%** | **67.1%** | 0.657 | 0.340 |
| CC | 80.5% | 51.7% | 82.1% | 55.9% | 0.685 | **0.348** |

Table 5: Quantitative comparison between our overlap-based methods and the state-of-the-art training-free method, Attend-and-Excite, on the T2I-CompBench where Stable Diffusion v2 is used as the backbone. The overlap-based approaches outperformed the baseline methods across all evaluation metrics.

| Method | TIFA Score (S) (↑) | TIFA Score (H) (↑) | BLIP-VQA Score (S) (↑) | BLIP-VQA Score (H) (↑) | Captioning Score (↑) | CLIP Similarity (↑) |
|---|---|---|---|---|---|---|
| Stable Diffusion | 84.8% | 71.3% | 86.7% | 74.6% | 0.767 | 0.325 |
| Attend-and-Excite | **95.6%** | **91.3%** | **95.8%** | **91.7%** | **0.769** | **0.333** |
| IoU | 96.8% | 93.8% | **97.7%** | **95.4%** | 0.789 | 0.331 |
| CoM Distance | **97.3%** | **94.6%** | **97.7%** | **95.4%** | **0.794** | **0.334** |
| CC | 96.0% | 92.1% | 97.1% | 94.2% | 0.788 | **0.334** |

**Background Entites.** While well-known benchmarks for visual compositional generation typically exclude background entities, we extended the scope of evaluation by conducting an experiment incorporating background elements. Specifically, we utilized the CoM distance as the loss function over 96 prompts structured in the format: "a {foreground entity} and a {background entity}". Stable Diffusion XL served as the backbone model, the foreground entity was chosen from the COCO dataset, and the background entity was varied across six categories: 'sea', 'beach', 'desert', 'jungle', 'sky', and 'mountain'. Table 6 presents the results from human assessments, revealing over 4% improvement in both harsh and soft evaluation scenarios.

Table 6: Quantitative comparison between the overlap-based method, CoM distance, and the state-of-the-art T2I model, Stable Diffusion XL, over prompts containing background entities such as 'sea', 'beach', 'desert', 'jungle', 'sky', and 'mountain' using soft and harsh human judgment.

| Method | Human Score (S) (↑) | Human Score (H) (↑) |
|---|---|---|
| Stable Diffusion | 91.7% | 89.6% |
| CoM Distance | **96.1%** | **94.8%** |

**Quality and Naturalness.** To further assess the quality of the generated images, we calculated the FID score for each method using the COCO Captions dataset (Lin et al., 2014). Table 7 presents the FID scores for the proposed overlap-based methods alongside various baseline models on two-entity and three-entity COCO prompts. Among the baselines, Structure Diffusion achieves the lowest FID scores—3.26 for two-entity prompts and 4.06 for three-entity prompts—indicating that it generates images with the most realistic overall appearance. However, our overlap-based optimization methods also demonstrate competitive FID scores. Notably, the KL divergence loss achieves the best results among our methods, with scores of 4.99 for two-entity prompts and 5.78 for three-entity prompts. Although these scores are slightly higher than those of Structure

Diffusion, they still indicate that our approach produces images of high quality and realism. Additionally, the FID scores for other overlap-based losses (IoU, CoM Distance, and CC) are significantly lower than those of the Stable Diffusion model and most other baselines, further highlighting the effectiveness of our method in preserving the quality of the generated images. Moreover, we adopted Coverage and Density (Naeem et al., 2020) metrics to assess the diversity and quality of the generated images, respectively. Table 7 demonstrates that, excluding Structure Diffusion, our overlap-based methods outperform the other baselines in terms of both Coverage and Density metrics. Notably, while Structure Diffusion achieved the highest performance in terms of quality and naturalness, there is a significant gap in compositional generation capabilities between this model and our proposed overlap-based methods, as shown in Tables 2 and 3. This demonstrates that the overlap-based approach offers a balanced trade-off between enhancing compositional generation and maintaining overall quality.

Table 7: FID scores on two-entity and three-entity COCO captions using Stable Diffusion v1.4 as the backbone for image generation. Lower FID scores indicate higher quality and more realistic generated images. Structure Diffusion achieves the lowest FID scores among the baselines, while our KL Divergence loss yields the best FID scores among the overlap-based optimization methods. Furthermore, in terms of the Coverage and Density metrics, the proposed overlap-based metrics outperform most of the baseline models, with only Structure Diffusion demonstrating superior performance.

| Method | FID Score (Two Entities) ($\downarrow$) | FID Score (Three Entities) ($\downarrow$) | Coverage ($\uparrow$) | Density ($\uparrow$) |
|---|---|---|---|---|
| Stable Diffusion | 7.85 | 8.09 | 0.84 | 0.66 |
| Attend-and-Excite | 6.57 | 7.54 | 0.80 | 0.65 |
| Divide-and-Bind | 7.21 | 7.40 | 0.86 | 0.72 |
| Predicated Diffusion | 6.27 | 7.50 | 0.70 | 0.70 |
| Structure Diffusion | **3.26** | **4.06** | **0.92** | **0.90** |
| IoU | 5.09 | 5.85 | 0.83 | 0.73 |
| CoM Distance | 6.21 | 6.39 | **0.85** | 0.67 |
| KL Divergence | **4.99** | **5.78** | 0.84 | **0.76** |
| CC | 5.44 | 6.36 | 0.81 | 0.72 |

**Robustness to Complex and Free-style Prompts.** To demonstrate the robustness of our overlap-based approach in handling complex, lengthy, and free-form prompts, we conducted additional experiments using four-entity prompts from the COCO-Comp dataset as well as free-form COCO captions (Lin et al., 2014). The results, provided in Appendices D.1 and D.2, highlight the effectiveness of our overlap-based methods, showing improvements of 12% and 9.6% in the harsh TIFA score for the COCO captions and four-entity prompts of COCO-Comp, respectively.

**Robustness to Different Backbones.** To further showcase the robustness of our approach across various Stable Diffusion backbones, we repeated the previous experiments using Stable Diffusion v2-base, Stable Diffusion v2, and Stable Diffusion XL. Our approach consistently outperformed the baselines across all backbones, underscoring its adaptability to different model sizes. Notably, we observed improvements of 14.9% with Stable Diffusion v2 and 14.2% with Stable Diffusion XL. Additional details on these experiments can be found in Appendix D.3.

**Combining Overlap-based Loss Functions with Other Metrics.** As discussed in Section 4.4, attention overlap exhibits the strongest negative correlation with the success rate of generating all entities, making it a key factor in the entity-missing problem. In addition to applying overlap-based loss functions ($IoU$, $D_{CoM}$, $D_{KL}$, and $CC$) independently, we also explored their combination with other loss terms, such as intensity ($Int$) and variance ($Var$), detailed in Sections 4.1 and 4.2, respectively. Appendix D.4 reports the results of these experiments on two-entity prompts from our COCO-Comp dataset using the Stable Diffusion v1.4 model as the backbone. Combining the overlap-based loss functions with intensity and variance provides

some additional improvements on certain metrics compared to the overlap-based loss functions alone, with the combination of CoM distance and intensity performing best overall. However, the gains from combining losses were insignificant, emphasizing the pivotal role of attention overlap in the entity missing problem. Specifically, utilizing $D_{CoM}$ as the sole loss function led to a 24.1% improvement over the baseline Stable Diffusion. However, when combined with the intensity-based loss function ($Int$), the additional improvement was only 0.8% compared to using $D_{CoM}$ alone.

**Other Visual Compositional Generation Failure Modes.** We further evaluated the effectiveness of the proposed overlap-based metrics in addressing other compositional generation challenges, including incorrect attribute binding, incorrect spatial relationship, and numeracy-related issues. For attribute binding and spatial relationship tasks, we used 100 randomly selected prompts from the color and 2D-spatial sections of the T2I-CompBench benchmark, respectively. To generate numeracy-related images, we utilized 100 prompts in the format "number entity and number entity", where the number tokens were one of 'one', 'two', 'three', or 'four', and the entity tokens were selected from the COCO dataset. The human assessment-based results presented in Table 8 were obtained using the CoM distance metric as the loss function, applied exclusively to entity-related attention maps, with Stable Diffusion v2 serving as the backbone. Notably, for numeracy-related problems, the mean absolute error (MAE) was calculated as the absolute difference between the ground truth number in the prompt and the number of generated entities in the image across all prompts. As expected, entity missing is a fundamental issue in visual compositional generation, and solely enhancing entity preservation can significantly contribute to mitigating other related challenges.

Table 8: Quantitative comparison between the overlap-based method, CoM distance, and the state-of-the-art T2I model, Stable Diffusion v2, over other visual compositional generation tasks: attribute binding, spatial relationship, and numeracy. All the results are based on harsh human assessments. For the attribute binding and spatial relationship tasks, the success rate of the method in accurately generating the corresponding attribute or relation is evaluated. In contrast, for the numeracy task, the mean absolute error (MAE) between the number of entities generated in the images and the ground truth number specified in the prompts is assessed.

| Method | Attribute Binding ($\uparrow$) | Spatial Relationship ($\uparrow$) | Numeracy ($\downarrow$) |
|---|---|---|---|
| Stable Diffusion | 59.75% | 56.33% | 2.74 |
| CoM Distance | **73.75%** | **63.66%** | **2.09** |

## 6.3 Qualitative Results

Figures 1 and 6 demonstrate qualitative comparison of images generated by our overlap-based methods and state-of-the-art baselines, Attend-and-Excite and Predicated Diffusion, on a diverse set of prompts from COCO-Comp dataset. As evident from the examples in Figure 6, Attend-and-Excite and Predicated Diffusion often struggle to generate all entities distinctly and well-separated. Instead, they tend to produce images where one entity is clearly rendered while others are poorly formed or missing entirely. For the prompt "a bicycle and a mirror", these models generate an image with only the mirror appearing well-defined, while the bicycle is poorly represented. Similarly, for "a bottle and a sheep," a clear bottle is generated, but the sheep entity is obscured or ill-defined. This issue becomes even more pronounced in three-entity scenarios. For example, given the prompt "boat, cow, and cup", both Attend-and-Excite and Predicated Diffusion completely fail to generate one of the three entities, omitting it entirely from the image.

In contrast, our proposed overlap-based methods consistently produce images that accurately capture the composition outlined in the input prompts. By minimizing the overlap between attention maps of distinct entities, our approach ensures that each subject is well-separated within the attention map space and correctly associated with its corresponding attributes. Further qualitative results using additional backbones (Stable Diffusion XL, Stable Diffusion v2, Stable Diffusion 1.4, Stable Diffusion v2-base) and datasets (COCO-Comp, COCO, and the prompts containing background entities) are provided in Appendix E.

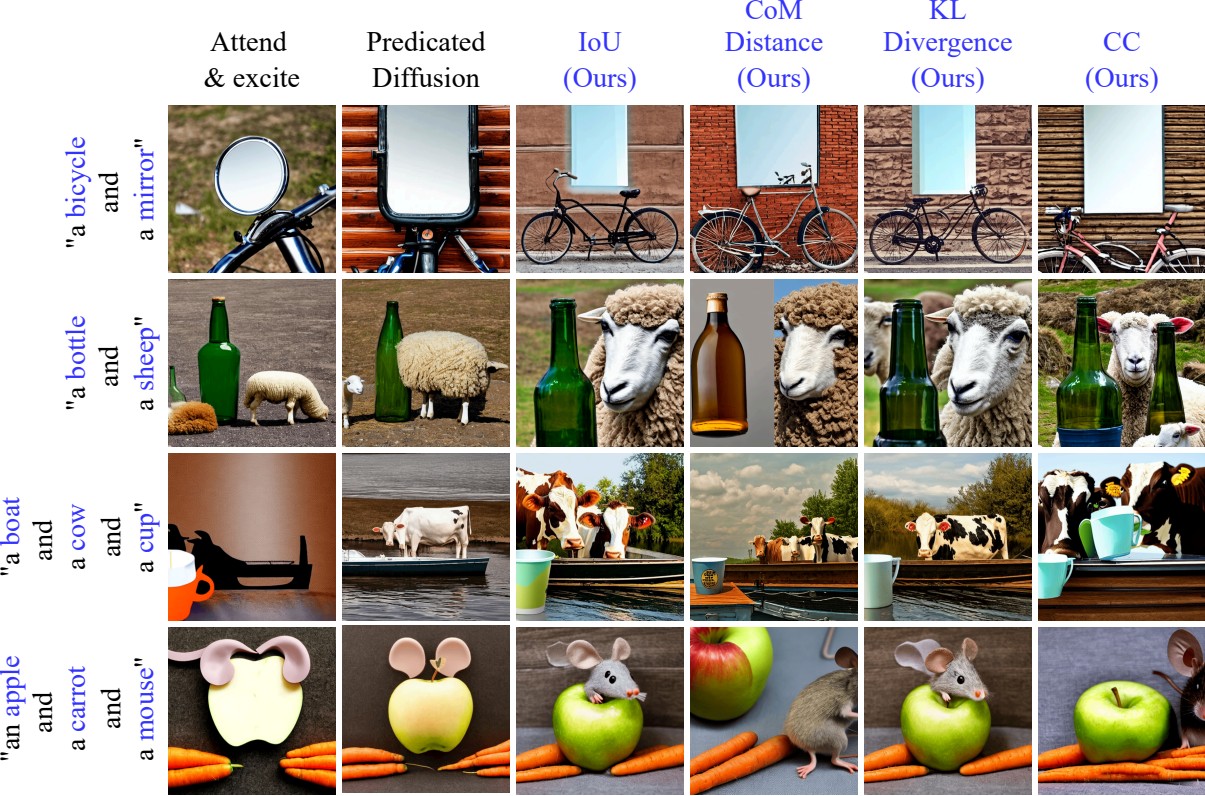

Figure 6: Qualitative comparison of images generated by our overlap-based methods (IoU, CoM Distance, KL Divergence, and CC) and state-of-the-art baselines (Attend-and-Excite and Predicated Diffusion) on various textual prompts. Our approach consistently generates images that faithfully reflect the composition specified in the prompts, with well-separated and correctly bound entities. In contrast, baselines like Attend-and-Excite and Predicated Diffusion often struggle to generate all entities distinctly, with some entities being poorly represented or missing entirely, especially in scenarios with three entities.

**Risk of Introducing Undesirable Biases.** It is important to clarify that reducing overlap between attention maps does not necessarily correlate with increasing spatial distance between entities in the generated image and, hence, generating unnatural images. Remarkably, three of our overlap-based methods (IoU, KL Divergence, and CC) only aim to reduce the overlap between attention maps of entities, not to push entities apart spatially. For instance, two entities could be placed side-by-side in the generated image while the IoU value between the corresponding attention maps is zero. As a real example, as illustrated in Figure 14 in Appendix F (the plate and pizza example), even when we reduce the overlap between attention maps during the image generation, the spatial relationship between entities (pizza inside the plate) is maintained, resulting in a realistic and natural image being generated.

## 7 Conclusion

Through this work, we conducted a comprehensive study of the entity missing problem, a critical compositional generation failure mode in T2I diffusion models. By investigating three hypothetical sources of the problem through the lens of cross-attention mechanisms - attention intensity, attention spread, and attention overlap - we identified the crucial role that overlap between attention maps of entities play in the entity missing problem. Building upon this insight, we proposed four overlap-based loss functions: $IoU$, $D_{CoM}$, $D_{KL}$, and $CC$, which can be applied over the latent embedding as training-free approaches to mitigate the entity missing

issue during denoising process. Extensive experiments on a diverse set of prompts from our proposed dataset COCO-Comp to free-style COCO captions and well-known compositional benchmarks T2I-Compbench and HRS-Bench, which all together span various linguistic structures and entity counts, demonstrated that our proposed methods significantly outperform state-of-the-art baselines in faithfully generating entities, as measured by commonly-used evaluation metrics such as human evaluation, visual question answering (VQA), image captioning score, and CLIP similarity. These results suggest that attention overlap is a primary factor contributing to the entity missing problem in T2I diffusion models. We also evaluate the quality and naturalness of the images generated by our overlap-based methods using the FID score, Coverage, and Density metrics. All metrics indicate that the degradation in quality and diversity is negligible. Furthermore, qualitative results suggest that the application of overlap-based loss functions during the denoising steps does not introduce undesirable biases, such as unnecessary and unnatural spatial distances between entities. While there are limitations to the proposed method (Appendix G), the proposed overlap-based loss functions provide an effective and flexible plug-and-play framework for addressing the entity missing problem without the need for expensive fine-tuning or retraining of the model. Furthermore, our experiments indicated that the exclusive enhancement of entity preservation can contribute to the alleviation of other challenges in compositional generation, such as incorrect attribute binding, incorrect spatial relationship, and issues related to numeracy. We believe this work paves the way for further research into training-free approaches for enhancing the compositional generation capabilities of T2I diffusion models, ultimately leading to more reliable and semantically coherent image synthesis.

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

# Appendix

## A    COCO-Comp Dataset

Unfortunately, many previous works have utilized datasets with limited diversity regarding the entities and categories represented. In response, we emphasize the importance of covering a diverse set of entities spanning a wide range of categories, including animals, foods, furniture, and more, by adopting 71 objects from the COCO dataset Lin et al. (2014) as our entity set. These diverse entities, detailed in table 9, can comprehensively assess the ability of the T2I model to encounter the complexity of real-world environments and even unseen combinations of entities, which is neglected in the previous works. Notably, we filtered out 20 entities from the 91 entities represented in the COCO dataset because they were two-part words, such as "wine glass" or plural, such as "scissors".

Adopting the 71 remaining objects from the COCO dataset, we generated our dataset called *COCO-Comp*, containing two groups of prompts, each designed to challenge the compositional generation capability of the T2I model. The first group comprised 2485 prompts featuring two entities, structured in the format of "a {entity} and a {entity}", while the second group consisted of 57155 prompts containing three entities, presented as "a {entity} and a {entity} and a {entity}". The number of prompts for different entity counts is shown in table 10. For each group, we divided the prompt set into two sets, a validation set for hyper-parameter finding and a test set for evaluation, in such a way that there is no common entity between these sets. This strict splitting strategy, neglected in previous works, ensures no data leakage during the testing phase. More details about the entities, prompts, and the splitting process are available in supplementary materials.

Notably, adopting these fixed formats in the COCO-Comp dataset serves at least two purposes. Firstly, it standardizes the presentation of prompts, ensuring consistency over all experiments and reducing the effect of other irrelevant factors that may implicitly contribute to the evaluation of the entity missing problem. Secondly, these fixed formats enable us to systematically explore a wide range of entity combinations, from common pairings to more unusual configurations. Including common and rare combinations evaluates the model's ability to generalize beyond familiar training data and adapt to novel scenarios. However, we also conducted experiments on the COCO captions to go beyond these fixed prompt formats.

Table 9: Entities and the corresponding categories of the COCO-Comp dataset, where black and blue ones are entities in the test and validation sets, respectively.

| Animal | Vehicle | Clothing | Digital Device | Etable | Home & Other |
|--------|---------|----------|----------------|--------|--------------|
| bird | motorcycle | shoe | tv | banana | spoon, bench |
| dog | boat | handbag | remote | apple | cup, kite |
| cat | airplane | tie | laptop | pizza | chair, frisbee |
| horse | bus | backpack | keyboard | cake | desk, snowboard |
| sheep | train | umbrella | | donut | blender, person |
| cow | truck | hat | | orange | knife, skateboard |
| elephant | bicycle | | | sandwich | toilet, surfboard |
| bear | car | | | broccoli | door, suitcase |
| zebra | | | | carrot | microwave, bottle |
| giraffe | | | | | plate |
| mouse | | | | | fork |
| | | | | | bowl |
| | | | | | couch |
| | | | | | bed |
| | | | | | mirror |
| | | | | | window |

Table 10: Number of prompts for different entity counts in the COCO-Comp dataset and COCO captions

| Entity Count | #Prompts in COCO-Comp | #COCO Captions |
|---|---|---|
| Two Entities (Validation) | 190 | - |
| Two Entities (Test) | 1275 | 1000 |
| Three Entities | 1000 | 1000 |

## B  Compositional Evaluation Metrics

Evaluating compositional generation failure modes, like entity missing, can be challenging due to ambiguity in interpreting a missing occurrence. Hence, we applied four methods, from human evaluation to VLM-based ones, to obtain a holistic evaluation framework for the entity missing problem.

**Human Score:** Given a prompt along with the corresponding generated image, we ask the evaluator to assign each entity a score from a discrete set $\{0, 0.5, 1\}$, where a score of 0 denotes the absence of the entity in the generated image, a score of 1 indicates that the entity is wholly and faithfully represented, and a score of 0.5 signifies partial observability of the entity within the image. Subsequently, we provide two options for aggregating the assigned scores across all entities in a prompt: soft and harsh averaging strategies. Under the soft evaluation paradigm, scores are aggregated using standard arithmetic averaging without imposing additional constraints. Conversely, in the harsh human evaluation, we adopt a more stringent criterion by treating entities deemed to be partially observable (assigned a score of 0.5) as though they are absent from the image (assigned a score of 0). Finally, all the scores across prompts are averaged normally.

**VQA Score:** Through the visual question answering (VQA) evaluation method, for each entity in a given pair of prompts and generated images, we prepare a question, "Is there any {entity} in the image?" and give it to the VQA model along with the corresponding image. If the VQA model answers "yes", we consider a score of 1; otherwise, it is 0. Similar to human evaluation, we proposed two options of soft and harsh averaging in such a way that in a harsh scenario, the overall score of a prompt is 1, if and only if the VQA answers to all questions are "yes". At the end, we average all the prompt scores. Notably, BLIP-VQA Li et al. (2022) and TIFA Hu et al. (2023) models are used for VQA evaluation.

**Captioning Score:** In caption evaluation, we use an image captioning model called BLIP-Captioner (Li et al., 2022) to produce a caption for each generated image. Then, the cosine similarity between the caption and the corresponding original prompt in CLIP (Radford et al., 2021) embedding space is considered as the score of that prompt, and the final score is calculated by averaging across prompt scores.

**CLIP Similarity:** In this evaluation method, we calculate the cosine similarity between a prompt and its corresponding generated image in CLIP's shared embedding space, treating it as the score (Hessel et al., 2021). Similar to other metrics, Averaging these scores across prompts yields the final score.

## C  Attention Map-based Loss Functions

### C.1  State-of-the-art Loss Functions

The proposed loss functions of three well-knwon training-free approach, Attant-and-Excite (Chefer et al., 2023), Divide-and-Bind (Li et al., 2023), and Predicated-Diffusion (Sueyoshi & Matsubara, 2024), are demonstrated in equations 8, 9, and 10, respectively. At each denoising step $t$, the latent embedding $z_t$ is modified in the direction opposite to the gradient of these loss functions with respect to $z_t$.

$$L_{Attent-and-Excite} = \max_{e \in \mathcal{E}} L_t^e \qquad \text{where} \qquad L_t^e = 1 - \max(A_t^e) \tag{8}$$

$$L_{Divide-and-Bind} = -\min_{e \in \mathcal{E}} \sum_{i,j} |A_t^e[i,j] - A_t^e[i+1,j]| + |A_t^e[i,j] - A_t^e[i,j+1]| \tag{9}$$

$$L_{Predicated-Diff} = -\sum_{e\in\mathcal{E}} \log\left(1 - \prod_{i,j}(1 - A_t^e[i,j])\right) \tag{10}$$

## C.2 Extended Version of Introduced Metrics

Equations 11 and 12 provide generalized formulations of $Int$ and $Var$ metrics to deal with more than two entities, respectively. The extended $Int$ metric 11 takes the minimum intensity across all entities, while the generalized $Var$ metric 12 averages the variance around the center of mass over all entities. Moreover, equations 13, 14, 15, and 16 are the extended version of the overlap-based metrics to handle prompts with multiple entities. The extended versions of IoU (equation 13) and KL divergence (equation 15) are computed as the average value over all entity pairs. The CoM distance (equation 14) considers the area of the polygon formed by the CoMs of all entities. The CC (equation 16) takes the maximum attention score at each location across all entity maps. Note that in equation 14, $T_N$ denotes the polygon formed by the $CoM$s of attention maps as the $N$ vertices.

$$Int(A_t^{e_1}, \ldots, A_t^{e_N}) = \min\{Int(A_t^{e_1}), \ldots, Int(A_t^{e_N})\} \tag{11}$$

$$Var(P_t^{e_1}, \ldots, P_t^{e_N}) = \frac{1}{N}\sum_{i=1}^{N} Var(P_t^{e_i}) \tag{12}$$

$$IoU(P_t^{e_1}, \ldots, P_t^{e_N}) = \frac{1}{\binom{N}{2}}\sum_{k=1}^{N}\sum_{l=k+1}^{N} IoU(P_t^{e_k}, P_t^{e_l}) \tag{13}$$

$$D_{CoM}(P_t^{e_1}, \ldots, P_t^{e_N}) = Area(T_N) \tag{14}$$

$$D_{KL}(P_t^{e_1}, \ldots, P_t^{e_N}) = \frac{1}{\binom{N}{2}}\sum_{k=1}^{N}\sum_{l=k+1}^{N} D_{KL}(P_t^{e_k}, P_t^{e_l}) \tag{15}$$

$$CC(P_t^{e_1}, \ldots, P_t^{e_N}) = \frac{1}{H\times W}\sum_{i,j}\max\{P_t^{e_1}[i,j], \ldots, P_t^{e_N}[i,j]\} \tag{16}$$

# D Additional Quantitative Results

## D.1 Results of More Complex Prompts

To evaluate the robustness of our overlap-based methods to prompts containing more entities, we conducted an experiment using four-entity prompts from the COCO-Comp dataset with Stable Diffusion v1.4 as the backbone. Table 11 demonstrates the superiority of our overlap-based methods over two strong baselines in dealing with complex prompts.

Table 11: Performance comparison between our overlap-based methods and state-of-the-art training-free approaches over the **four-entity** prompts from COCO-Comp dataset, using the Stable Diffusion v1.4 as the backbone model. The proposed overlap-based methods consistently outperform the baselines.

| Method | TIFA Score (S) (↑) | TIFA Score (H) (↑) | BLIP-VQA Score (S) (↑) | BLIP-VQA Score (H) (↑) | Captioning Score (↑) | CLIP Similarity (↑) |
|---|---|---|---|---|---|---|
| Attend-and-Excite | **70.8%** | **23.2%** | **82.0%** | **45.4%** | 0.738 | **0.322** |
| Predicated Diffusion | 65.4% | 8.8% | 70.6% | 13.4% | **0.760** | 0.320 |
| IoU | 77.2% | **32.8%** | 84.3% | **47.8%** | **0.757** | **0.329** |
| CoM Distance | **78.2%** | 28.2% | **85.2%** | 45.4% | 0.744 | 0.318 |

## D.2 Results of Free-Style Prompts

In the previous experiments, prompts followed a fixed format of "a {entity} and a {entity}". To evaluate the generalizability of our method to more diverse linguistic structures, we also tested it on free-form captions from the COCO dataset (Lin et al., 2014). Table 12 presents the results for COCO captions containing at least three entities, utilizing Stable Diffusion 1.4 as the backbone model. The overlap-based loss functions demonstrate significant improvements over the baseline methods.

Table 12: Performance comparison between our overlap-based methods and state-of-the-art training-free approaches over the COCO captions with at least three entities, using the Stable Diffusion v1.4 as the backbone model. The proposed overlap-based methods consistently outperform the baselines, with the IoU achieving the best results in VQA-based evaluations.

| Method | TIFA Score (S) (↑) | TIFA Score (H) (↑) | BLIP-VQA Score (S) (↑) | BLIP-VQA Score (H) (↑) | Captioning Score (↑) | CLIP Similarity (↑) |
|---|---|---|---|---|---|---|
| Stable Diffusion | 67.0% | 22.2% | 70.3% | 28.5% | 0.781 | 0.313 |
| Attend-and-Excite | **85.7%** | **62.1%** | **90.3%** | **73.2%** | 0.816 | 0.314 |
| Divide-and-Bind | 84.0% | 58.3% | 88.3% | 68.6% | **0.818** | **0.315** |
| Structure Diffusion | 79.1% | 46.9% | 83.0% | 55.0% | 0.816 | 0.310 |
| Predicated Diffusion | 79.2% | 45.6% | 81.6% | 51.4% | 0.812 | 0.312 |
| IoU | **90.3%** | **74.1%** | **93.6%** | **81.8%** | 0.821 | 0.317 |
| CoM Distance | 88.7% | 69.4% | 92.5% | 80.0% | 0.820 | 0.314 |
| KL Divergence | 89.5% | 71.9% | 92.6% | 79.2% | **0.824** | **0.318** |
| CC | 89.7% | 73.2% | 93.2% | 81.5% | 0.812 | 0.313 |

## D.3 Results of Other Backbones

In addition to SD v1.4, we assessed our approach on other recent backbones, **SD v2-base**, **SD v2** (Rombach et al., 2022), and **SD-XL** (Podell et al., 2023), to validate its effectiveness across different model sizes. Tables 13 and 14 present the results on two-entity and three-entity prompts from our COCO-Comp dataset using Stable Diffusion v2-base. Moreover, tables 15 and 16 demonstrate the obtained results on two-entity prompts of the COCO-Comp dataset, using Stable Diffusion v2 and Stable Diffusion XL, respectively. Across all the backbones, the overlap-based methods outperform other baselines in all evaluation metrics. This suggests that our overlap-based approach can generalize well to different diffusion model architectures and sizes.

Table 13: Performance comparison between our overlap-based methods and state-of-the-art training-free approaches using the **two-entity** prompts from the COCO-Comp dataset on the **Stable Diffusion v2-base** model as the backbone. The proposed overlap-based methods significantly outperform the baselines across all evaluation metrics, with the CoM Distance and CC methods achieving the highest scores.

| Method | TIFA Score (S) (↑) | TIFA Score (H) (↑) | BLIP-VQA Score (S) (↑) | BLIP-VQA Score (H) (↑) | Captioning Score (↑) | CLIP Similarity (↑) |
|---|---|---|---|---|---|---|
| Stable Diffusion | 76.0% | 52.3% | 78.8% | 57.8% | 0.794 | 0.305 |
| Attend-and-Excite | **82.9%** | **66.0%** | **85.3%** | **70.7%** | **0.805** | **0.308** |
| Divide-and-Bind | 77.9% | 56.0% | 80.3% | 69.9% | 0.802 | 0.307 |
| Predicated Diffusion | 70.8% | 47.0% | 78.0% | 58.6% | 0.758 | 0.285 |
| Composable Diffusion | 64.4% | 31.7% | 72.6% | 46.3% | 0.772 | 0.301 |
| IoU | 96.2% | 92.4% | 97.8% | 95.7% | 0.841 | 0.325 |
| CoM Distance | **96.6%** | **93.3%** | 97.7% | 95.5% | **0.844** | **0.326** |
| KL Divergence | 96.3% | 92.8% | 98.0% | 96.0% | 0.837 | 0.325 |
| CC | 96.2% | 92.6% | **98.2%** | **96.5%** | 0.838 | 0.323 |

Table 14: Performance comparison between our overlap-based methods and state-of-the-art training-free approaches using the **three-entity** prompts from the COCO-Comp dataset on the **Stable Diffusion v2-base** model as the backbone. The proposed overlap-based methods significantly outperform the baselines across all evaluation metrics, with the CoM Distance method achieving the highest scores.

| Method | TIFA Score (S) (↑) | TIFA Score (H) (↑) | BLIP-VQA Score (S) (↑) | BLIP-VQA Score (H) (↑) | Captioning Score (↑) | CLIP Similarity (↑) |
|---|---|---|---|---|---|---|
| Stable Diffusion | 67.0% | 22.2% | 70.3% | 28.5% | 0.781 | 0.313 |
| Attend-and-Excite | **70.0%** | **27.2%** | **73.8%** | **33.8%** | **0.785** | **0.314** |
| Divide-and-Bind | 67.7% | 24.0% | 71.2% | 29.7% | 0.742 | **0.314** |
| Predicated Diffusion | 62.0% | 19.5% | 70.6% | 30.5% | 0.730 | 0.292 |
| Composable Diffusion | 46.8% | 5.1% | 59.9% | 17.9% | 0.722 | 0.291 |
| IoU | 91.9% | 76.7% | 94.7% | 84.3% | 0.819 | 0.340 |
| CoM Distance | **94.5%** | **85.3%** | **96.8%** | **91.2%** | **0.823** | **0.343** |
| KL Divergence | 89.2% | 69.3% | 0.92.2% | 77.4% | 0.819 | 0.340 |
| CC | 94.0% | 82.6% | 96.7% | 90.3% | 0.819 | 0.341 |

Table 15: Performance comparison between our overlap-based methods and the state-of-the-art training-free method, Attend-and-Excite, using the **two-entity** prompts from the COCO-Comp dataset on the **Stable Diffusion v2** model as the backbone. The proposed overlap-based methods significantly outperform the baselines across all evaluation metrics, with the CoM Distance method achieving the highest scores.

| Method | TIFA Score (S) (↑) | TIFA Score (H) (↑) | BLIP-VQA Score (S) (↑) | BLIP-VQA Score (H) (↑) | Captioning Score (↑) | CLIP Similarity (↑) |
|---|---|---|---|---|---|---|
| Stable Diffusion | 81.4% | 63.2% | 83.3% | 66.7% | 0.818 | 0.308 |
| Attend-and-Excite | **90.1%** | **80.2%** | **92.7%** | **85.5%** | **0.824** | **0.314** |
| IoU | 93.9% | 91.4% | 98.3% | 96.6% | **0.842** | 0.323 |
| CoM Distance | **97.5%** | **95.1%** | **98.9%** | **97.7%** | 0.841 | **0.325** |
| CC | 96.0% | 92.1% | 98.0% | 95.9% | **0.842** | 0.324 |

Table 16: Performance comparison between our overlap-based methods and the strong baseline model, **Stable Diffusion XL**, using the **two-entity** prompts from the COCO-Comp dataset. The proposed overlap-based methods significantly outperform the baselines across all evaluation metrics, with the CoM Distance method achieving the highest scores.

| Method | TIFA Score (S) (↑) | TIFA Score (H) (↑) | BLIP-VQA Score (S) (↑) | BLIP-VQA Score (H) (↑) | Captioning Score (↑) | CLIP Similarity (↑) |
|---|---|---|---|---|---|---|
| Stable Diffusion | 88.0% | 76.2% | 88.6% | 77.6% | 0.818 | 0.315 |
| CoM Distance | **95.2%** | **90.4%** | **96.0%** | **91.9%** | 0.826 | **0.322** |
| KL Divergence | 91.1% | 82.4% | 92.0% | 84.3% | **0.827** | **0.322** |
| CC | 90.2% | 80.4% | 90.9% | 82.0% | 0.820 | 0.320 |

## D.4  Results of Combined Methods

Table 17 demonstrates the quantitative results of combining overlap-based methods with other loss functions ($Int$ and $Var$) on the two-entity prompts from our dataset COCO-Comp using the Stable Diffusion v1.4 model as the backbone. The improvement induced by this combination is relatively low, proposing the attention overlap as the root cause of the entity missing problem.

Table 17: Performance comparison of overlap-based metrics and their combinations with variance and intensity loss functions on the two-entity prompts of the COCO-Comp dataset using the Stable Diffusion v1.4 model as the backbone. The proposed overlap-based methods and their combination with non-overlap-based ones consistently outperform the baselines across all evaluation metrics. While incorporating intensity and variance-based loss functions alongside overlap-based ones generally enhances the performance of compositional generation, the extent of this improvement is minimal, suggesting that attention overlap remains the primary factor contributing to the issue of missing entities.

| Method | TIFA Score (S) (↑) | TIFA Score (H) (↑) | BLIP-VQA Score (S) (↑) | BLIP-VQA Score (H) (↑) | Captioning Score (↑) | CLIP Similarity (↑) |
|---|---|---|---|---|---|---|
| Stable Diffusion | 69.9% | 40.0% | 80.0% | 44.2% | 0.789 | 0.297 |
| Attend-and-Excite | 87.6% | 76.5% | 92.2% | 84.9% | 0.811 | 0.312 |
| Divide-and-Bind | 86.5% | 73.6% | 91.9% | 83.9% | 0.815 | 0.309 |
| Structure Diff | 74.3% | 49.6% | 77.2% | 54.9% | 0.792 | 0.299 |
| Predicated Diff | **90.6%** | **81.6%** | **92.7%** | **85.7%** | **0.823** | **0.314** |
| Composable Diff | 66.2% | 37.6% | 73.1% | 49.3% | 0.761 | 0.297 |
| IoU | 92.7% | 85.3% | 95.0% | 90.1% | 0.824 | 0.317 |
| CoM Distance | **94.0%** | **88.2%** | **95.6%** | **91.3%** | **0.825** | **0.318** |
| KL Divergence | 91.7% | 83.5% | 94.5% | 88.9% | 0.818 | 0.316 |
| CC | 92.2% | 84.6% | 95.1% | 90.3% | 0.821 | 0.315 |
| IoU & Int | 93.8% | 87.7% | 96.5% | 92.9% | **_0.831_** | 0.319 |
| CoM Distance & Int | _94.8%_ | _89.8%_ | _96.7%_ | _93.5%_ | 0.825 | **0.319** |
| KL Divergence & Int | 91.7% | 83.7% | 94.8% | 89.7% | 0.814 | 0.314 |
| CC & Int | 92.5% | 85.2% | 96.3% | 92.8% | 0.824 | 0.316 |
| IoU & Var | 91.4% | 82.9% | 94.3% | 88.7% | 0.822 | **_0.321_** |
| CoM Distance & Var | **94.4%** | **89.0%** | **96.0%** | **92.2%** | 0.825 | _0.321_ |
| KL Divergence & Var | 90.8% | 81.6% | 94.5% | 89.1% | 0.813 | 0.315 |
| CC & Var | 91.5% | 83.4% | 93.9% | 88.0% | 0.813 | 0.320 |

## E  Additional Qualitative Results

Figures 7, 8, 9, 10, 11, and 12 present additional qualitative comparisons of images generated between our overlap-based methods and various baselines. More specifically, figures 7, 10, 11, and 12 contain

two-object prompts from our COCO-Comp dataset using **Stable Diffusion XL**, **Stable Diffusion v2**, **Stable Diffusion v1.4**, **Stable Diffusion v2-base**, respectively. Moreover, figure 8 includes free-style and complex prompts from the COCO dataset. Finally, figure 9 demonstrates the generated images from prompts containing background entities such as 'sea', 'beach', 'desert', and 'mountain'.

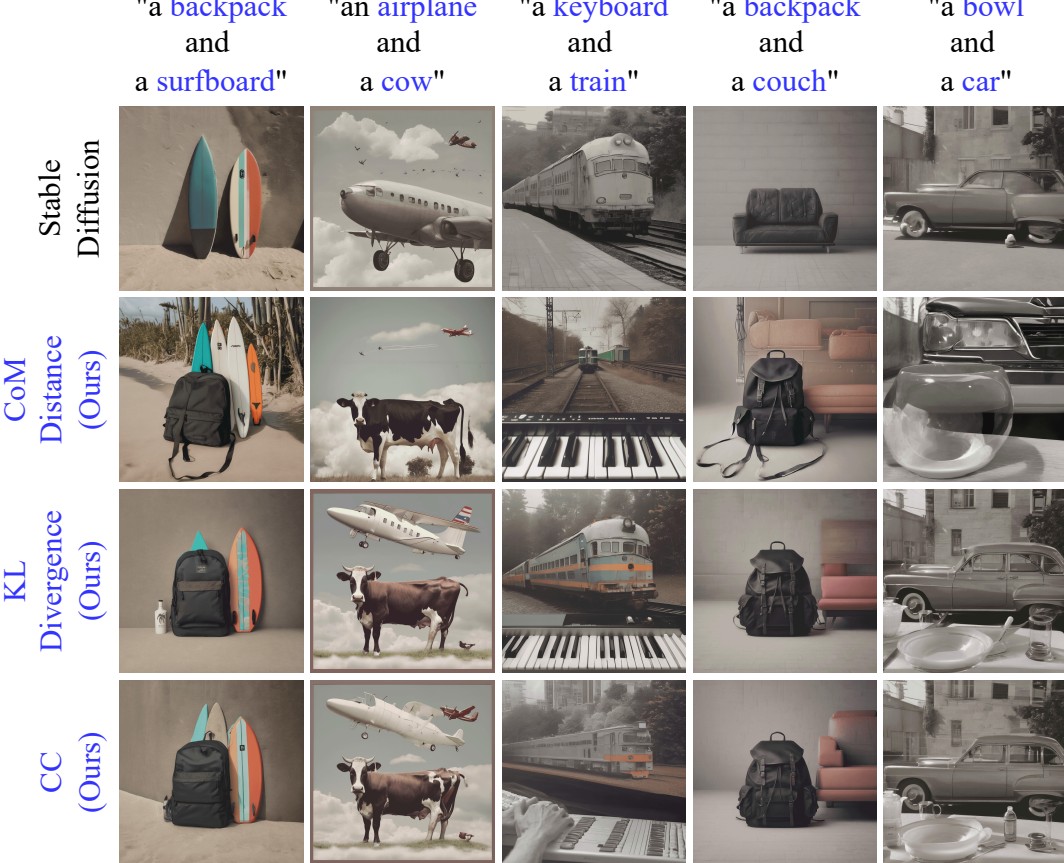

Figure 7: Qualitative comparison between our proposed overlap-based methods and the state-of-the-art baseline, Stable Diffusion XL, on diverse prompts from our COCO-Comp dataset.

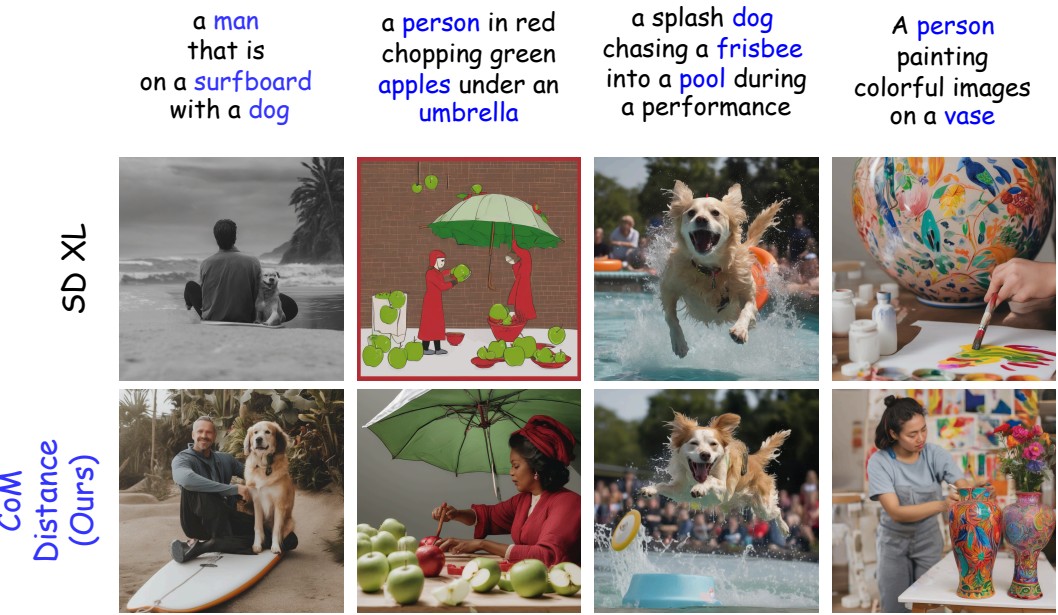

Figure 8: Qualitative comparison between our proposed overlap-based method, CoM distance, and the state-of-the-art baseline, Stable Diffusion XL, on free-style and complex prompts from the COCO dataset.

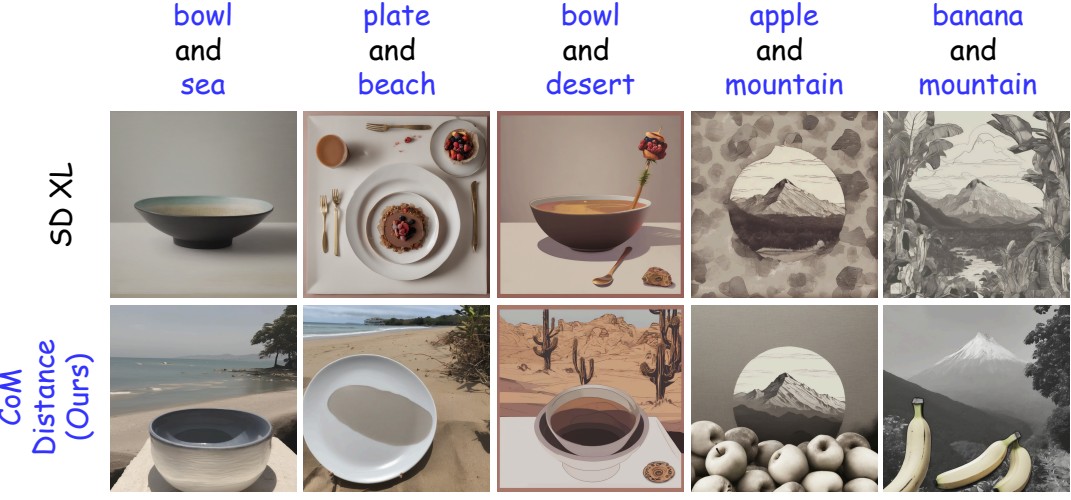

Figure 9: Qualitative comparison between our proposed overlap-based method, CoM distance, and the state-of-the-art baseline, Stable Diffusion XL, on prompts containing background entities such as 'sea', 'beach', 'desert', and 'mountain'.

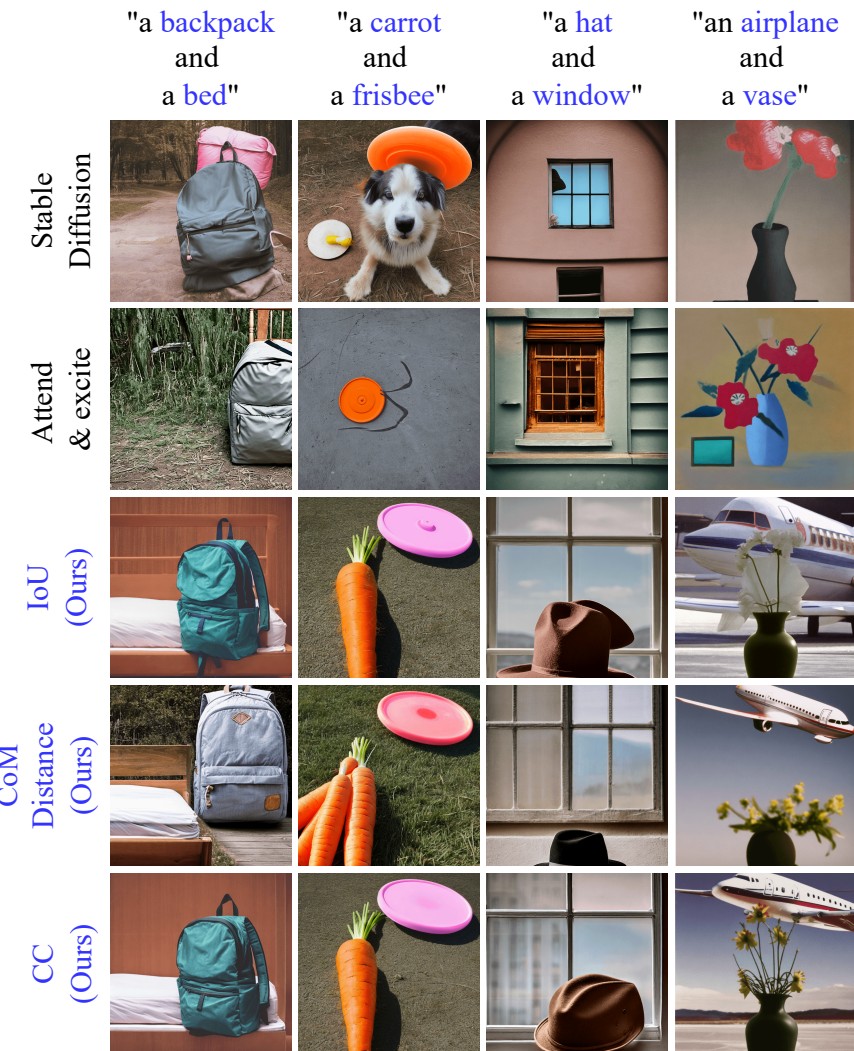

Figure 10: Qualitative comparison between all proposed overlap-based methods and the state-of-the-art baselines on diverse prompts from our COCO-Comp dataset using Stable Diffusion v2.

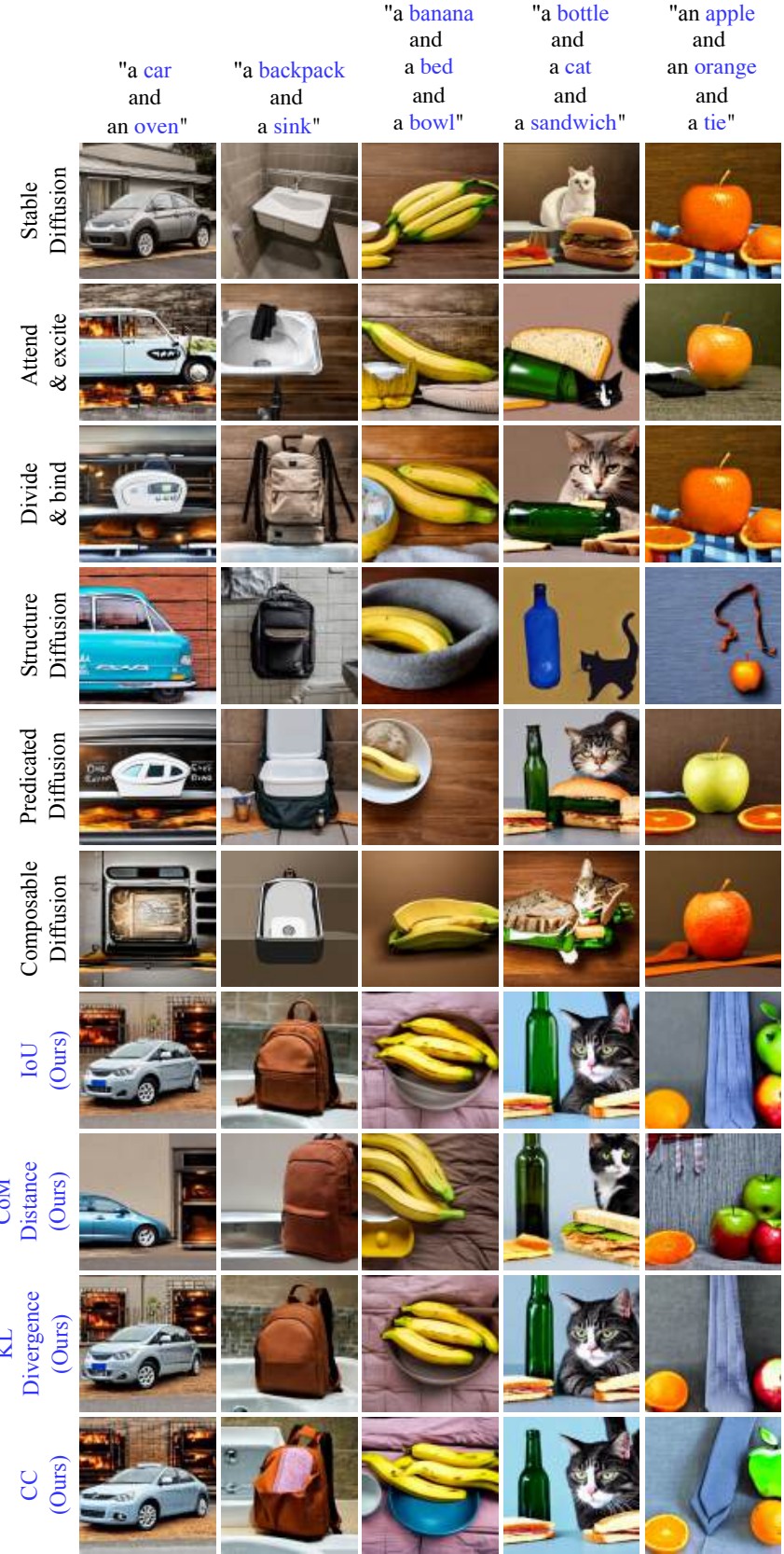

Figure 11: Qualitative comparison between all proposed overlap-based methods and the state-of-the-art baselines on diverse prompts from our COCO-Comp dataset using Stable Diffusion v1.4.

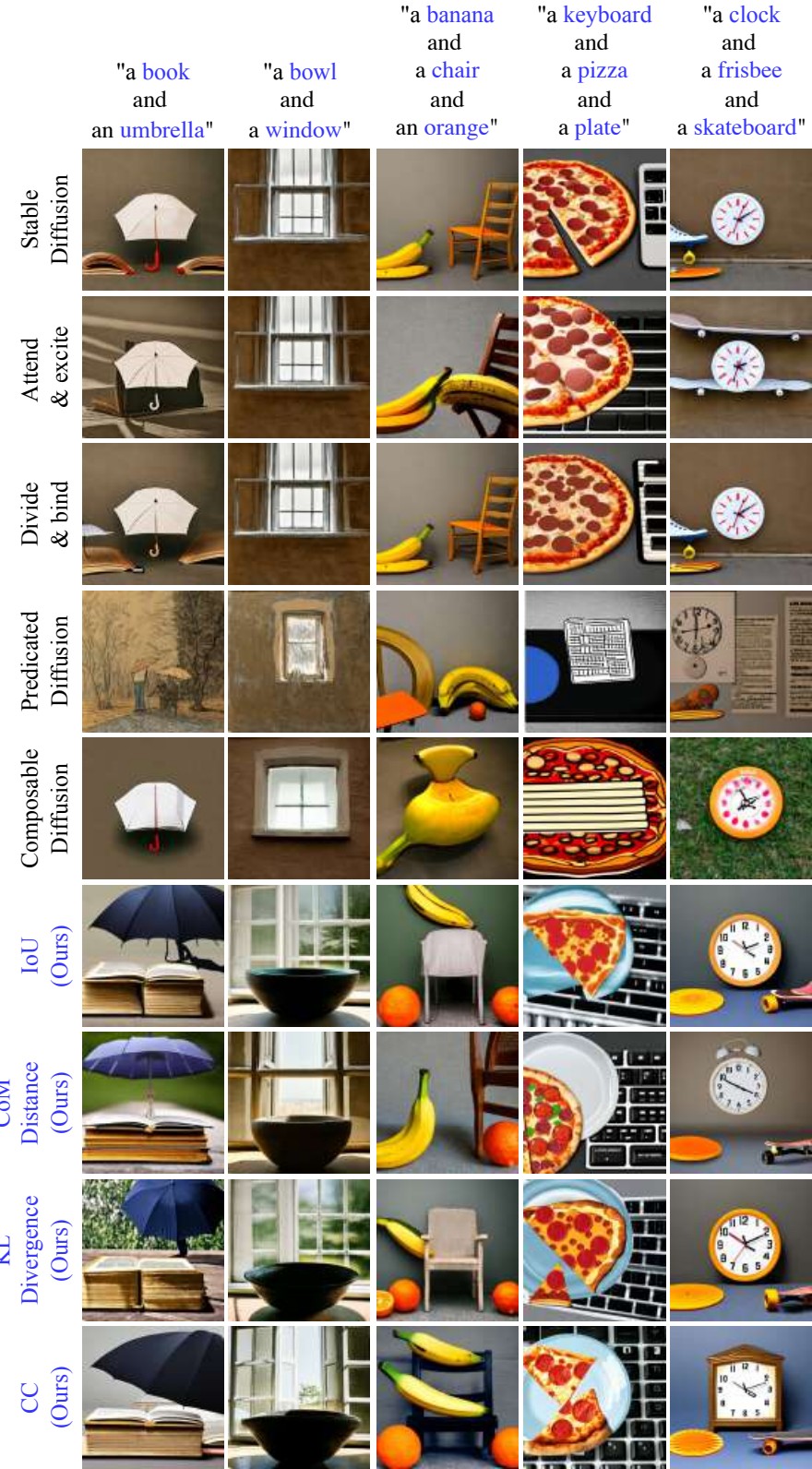

Figure 12: Qualitative comparison between all proposed overlap-based methods and the state-of-the-art baselines on diverse prompts from our COCO-Comp dataset using Stable Diffusion v2-base.

# F  Attention Map Evolution

Figure 13 provides insights into how the cross-attention maps evolve during the denoising process for different models, including our overlap-based methods. The figure depicts the attention maps for two entities at different denoising time steps (t=1, 3, 6, 10, 30, 50). In the initial time steps, the attention maps for the two entities are the same across all models because the random seed is fixed. This ensures a fair comparison by starting from the same initial attention distribution. As the denoising progresses, the attention maps evolve differently for each model. In the baseline models the attention maps remain largely overlapping and diffuse throughout the denoising process. In contrast, when using our overlap-based optimization method, the attention maps become more distinct and localized as the denoising steps proceed. Our approach effectively guides the diffusion model to minimize the overlap between attention maps of different entities, encouraging a clearer separation of entity representations. By the final time step, the attention maps for the two entities are largely non-overlapping and concentrated on the corresponding entities in the image.

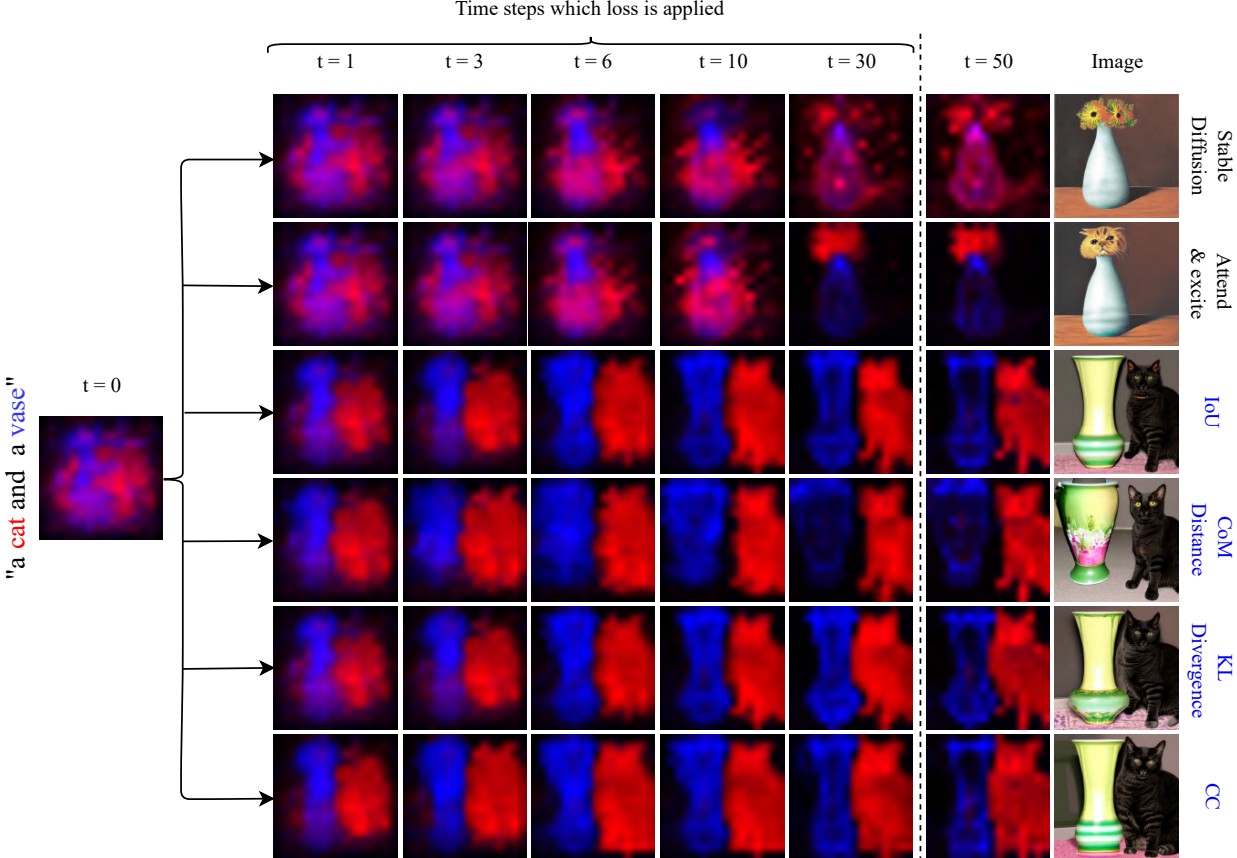

Figure 13: Comparison of attention maps during the denoising process for different training-free methods, including the proposed overlap-based methods. The attention maps for two entities are shown at various denoising time steps (t=1, 3, 6, 10, 30, 50). While the initial attention maps are identical across models due to fixed random seed, the proposed overlap-based methods ($IoU$, $D_{CoM}$, $D_{KL}$, $CC$) effectively minimizes overlap and encourages distinct, localized attention for each entity as denoising progresses, resulting in improved compositional alignment.

Moreover

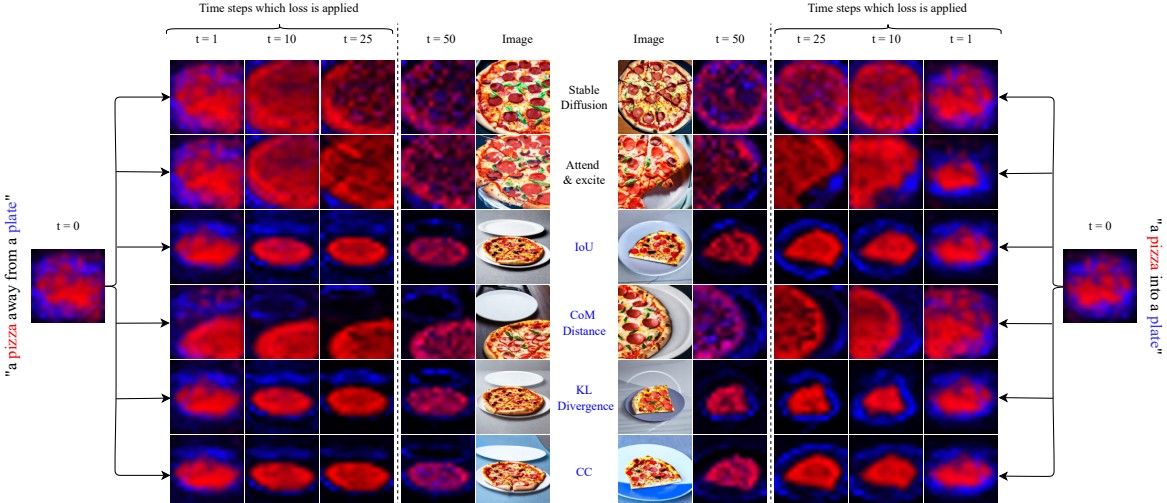

Figure 14: Comparison of attention maps during the denoising process for different training-free methods, including the proposed overlap-based methods. As demonstrated for the prompt "a pizza into a plate", applying our overlap-based methods does not push the entities (pizza and plate) far away from each other and, hence, generate unnatural images. On the other hand, when dealing with the prompt "a pizza away from a plate", our methods generate images entirely faithful to the input prompt. At the same time, Stable Diffusion and Attend-and-Excite fail to generate one entity (plate).

## G  Limitations

While our proposed training-free method offers a computationally efficient approach to mitigate compositional generation failure modes in T2I models, there are several limitations that should be acknowledged and addressed in the future research.

**Scalability limitations of training-free methods.**  Training-free methods may not be the ultimate solution for compositional generation failure modes, particularly when dealing with highly complex text prompts containing numerous entities. As the complexity and length of the prompts increase, these methods might struggle to effectively capture and represent the intricate relationships and dependencies between the entities. In such scenarios, extensive fine-tuning of the T2I model on carefully curated datasets tailored to the specific domain or task may be necessary to achieve optimal performance. However, this fine-tuning process often requires substantial computational resources and time, which could be a significant limitation for certain applications or users with limited computing power.

**Limitations of the CLIP text encoder.**  The poor capability of the CLIP text encoder (Radford et al., 2021) in understanding and representing the compositionality of complex texts (Yuksekgonul et al., 2023; Thrush et al., 2022) may have a significant impact on the performance of T2I models. Since most of these models rely on the CLIP text encoder as their backbone for processing textual inputs, any limitations or weaknesses in the encoder's ability to capture and encode the compositional structure and semantics of the text can propagate through the entire system, leading to suboptimal results or failure modes in the generated images. Surprisingly, most of the related works have not paid sufficient attention to this potential root cause of the failures. Instead, they often focus on proposing solutions without thoroughly investigating the underlying reasons behind the problems. They tend to attribute the issues to other factors without considering the limitations of the CLIP text encoder itself. To address this, we plan to focus our future work on investigating and improving the text encoding capabilities of T2I models, specifically targeting their ability to handle compositional complexities in text prompts.

**Limitations of commonly used evaluation metrics.** The most commonly used evaluation metrics for assessing the performance of T2I models, such as VQA score, CLIP similarity, and captioning score, heavily rely on Vision-Language Models (VLMs) or multi-modal generative models. While these metrics provide useful insights, they inherently suffer from the limitations and biases of the underlying models, particularly in terms of their ability to understand and evaluate compositional aspects of the generated images. These models often struggle to accurately capture and assess the fine-grained details, spatial relationships, and semantic coherence between different entities in the generated images. As a result, the evaluation scores obtained using these metrics may not always align with human perception and judgment of the compositional quality of the generated images. To mitigate this limitation and obtain a more reliable assessment of our proposed method, we conducted extensive human evaluation studies, which provide a more direct and intuitive measure of the compositional alignment between the input text and the generated images.

