# OpenReview forum: "Attention Overlap Is Responsible for The Entity Missing Problem in Text-to-image Diffusion Models!"
_TMLR — Accepted by TMLR_

### Review · Reviewer_jij7 · 2024-12-09

**Summary Of Contributions:**

1. The authors conduct a systematic analysis into three possible reasons of entity
  missing in T2I diffusion models.
2. Their investigation reveals the overlap between attention maps of entities as the primary factor.
3. Based on this finding, the authors design four criteria to measure the overlap of the attention maps of the entities. Then an inference time optimization is performed to promote less overlap.
4. Extensive experiments showcase the effectiveness of the proposed method.

**Audience:**

Yes

**Claims And Evidence:**

Yes

**Requested Changes:**

1. It is beneficial to include some discussion about applying all three metrics, i.e., the attention intensity, the attention spread, and the attention overlap, in the inference time optimization. From the result in Figure 5, the attention intensity and spread also seem to affect the alignment. The question is whether they are minor causes of the entity missing problem. Does applying them also contribute to the text-image alignment?

2. It is better to include the related work mentioned in *Strengths And Weaknesses* part.
3. Adding test cases of some background entities can better demonstrate the effectiveness of the proposed method. Some example background entities are: mountain, sea, beach, and sky.
4. The authors need to discuss the limitation of the proposed method of the entity type, especially the applicability of abstract or background entities.

**Strengths And Weaknesses:**

**Strengthes:**

1. Comprehensive validation experiments to confirm the cause of the entity missing problem.
2. All the four metrics depicting the attention overlap bring significant improvement, proving the claim that the attention overlap is the primary cause of entity missing.
3. Comprehensive experiments across different benchmarks, prompt types, and backbones.

**Weaknesses:**

1. The applicability of the proposed method is limited. The text-image alignment problem includes various aspects, while the entity missing problem is only part of it. The proposed method cannot handle many other problems like the alignment of the entity attribute, the numerical alignment, and the relation alignment.

2. Lacking related work. The related work mostly introduce methods with inference time optimization. However, methods to address these misalignment issues in the training phase is missing [1,2].
3. The entities listed in Table 7 are all foreground objects. The paper does not include test cases for background entities like sea, forest, or sky. Also, the method also cannot handle abstract entities like music or atmosphere.

[1] Sun, Jiao, et al. "Dreamsync: Aligning text-to-image generation with image understanding feedback." *Synthetic Data for Computer Vision Workshop@ CVPR 2024*. 2023.

[2] Jiang, Dongzhi, et al. "CoMat: Aligning Text-to-Image Diffusion Model with Image-to-Text Concept Matching." *arXiv preprint arXiv:2404.03653* (2024).

---

> ### Author Response · Authors · 2025-01-05
> **First Part of the Answer**
>
> We sincerely thank you for your insightful feedback and valuable recognition of our comprehensive validation experiments. Furthermore, we greatly appreciate your acknowledgment of the significant improvements brought by the overlap-based metrics and your positive assessment of our experiments across various benchmarks, prompt types, and model backbones.
>
> We would like to address your concerns and questions:
>
> **Issue:** The applicability of the proposed method is limited. The text-image alignment problem includes various aspects, while the entity missing problem is only part of it. The proposed method cannot handle many other problems like the alignment of the entity attribute, the numerical alignment, and the relation alignment.
>
> **Answer:** While our primary focus is on mitigating the issue of entity missing, we have expanded our evaluations to encompass broader aspects of text-to-image alignment, including spatial relationships, attribute binding, and numeracy-related challenges. These results are presented in Table 8 of the revised manuscript.
>
> Our findings indicate that solely improving entity preservation can effectively contribute to addressing other compositional generation challenges, thereby enhancing overall compositional alignment. More specifically, using the CoM distance metric, improvements of approximately **14%** and **7%** are achieved for the attribute binding and spatial relationship cases, respectively. Furthermore, a reduction of **0.65** in the mean absolute error (MAE) metric is observed for the counting problem.
>
> **Issue:** Lacking related work. The related work mostly introduces methods with inference time optimization. However, methods to address these misalignment issues in the training phase are missing [1,2].
>
> **Answer:** In response, we have incorporated several fine-tuning-based methods into the Related Work section of the revised manuscript (first paragraph of Section 2.2), including those mentioned by the reviewer, to ensure a more comprehensive and thorough discussion.
>
> **Issue:** The entities listed in Table 7 are all foreground objects. The paper does not include test cases for background entities like sea, forest, or sky. Also, the method also cannot handle abstract entities like music or atmosphere.
>
> **Answer:** We would like to clarify two key points:
>
> 1. **Background Entities:** We have extended our work to evaluate the effectiveness of our overlap-based methods on background-foreground compositions, as presented in Table 6 of the revised manuscript. These experiments demonstrate that our method remains effective even when applied to background entities such as 'sea', 'beach', 'desert', 'jungle', 'sky', and 'mountain'. More specifically, improvements of **4.4%** and **5.2%** were achieved when applying the CoM distance metric using Stable Diffusion XL as the backbone model, according to soft and harsh human assessments, respectively.
> 2. **Abstract Concepts:** The entity-missing problem specifically pertains to physical objects that can be visibly present or absent in an image. Abstract concepts, such as music or atmosphere, influence the entire image space and contribute to the overall scene, rather than being directly associated with the presence or absence of a specific object. As such, they do not fall within the category of entity missing. Moreover, any approach aimed at improving the alignment of abstract concepts in T2I models can be seamlessly integrated with our proposed method for object preservation.

---

> ### Author Response · Authors · 2025-01-05
> **Second Part of the Answer**
>
> **Requested Change:** It is beneficial to include some discussion about applying all three metrics, i.e., the attention intensity, the attention spread, and the attention overlap, in the inference time optimization. From the result in Figure 5, the attention intensity and spread also seem to affect the alignment. The question is whether they are minor causes of the entity missing problem. Does applying them also contribute to the text-image alignment?
>
> **Answer:** We believe that the requested change is addressed in Section D4 of the Appendix, where multiple combinations of the factors in question are assessed. As demonstrated in Table 17 of the revised manuscript, while all combinations show improvements in entity preservation, the overlap-based metrics play the most significant role. If you request other combinations of metrics, we can provide them in the final version of the manuscript.
>
> **Requested Change:** It is better to include the related work mentioned in Strengths And Weaknesses part.
>
> **Answer:** We have incorporated the mentioned related works into Section 2.2 of the revised manuscript.
>
> **Requested Change:** Adding test cases of some background entities can better demonstrate the effectiveness of the proposed method. Some example background entities are: mountain, sea, beach, and sky.
>
> **Answer:** As requested, we have added a new figure (Figure 9 in the revised manuscript) containing the background entities, using Stable Diffusion XL as the backbone.
>
> **Requested Change:** The authors need to discuss the limitation of the proposed method of the entity type, especially the applicability of abstract or background entities.
>
> **Answer:** As previously discussed, to address the concern regarding background entities, we conducted new experiments, and the results are presented in Table 6 and Figure 9 of the revised manuscript. However, abstract concepts contribute to the overall scene rather than directly influencing the presence or absence of a specific entity. Therefore, they do not fall within the scope of the entity-missing problem.
>
> We value your suggestions and hope this clarification provides a deeper understanding of the broader impact and applicability of our work.

---

### Review · Reviewer_q34m · 2024-12-09

**Summary Of Contributions:**

In this paper, the authors investigate the entity missing problem in text-to-image diffusion models, introducing a systematic analysis of three potential causes: insufficient attention intensity, excessive attention spread, and overlapping attention maps between entities. To address these challenges, the authors propose overlap-based loss functions that optimize cross-attention during generation, enabling better integration of multiple entities while maintaining compositional fidelity and reducing conflicts in visual representation.

**Audience:**

Yes

**Broader Impact Concerns:**

No broader impact concerns.

**Claims And Evidence:**

Yes

**Requested Changes:**

In the Appendix, the authors have presented experimental results on free-style prompts, which show promising performance. I suggest including some qualitative results as well to provide a clearer visualization of the model's effectiveness in these scenarios.

**Strengths And Weaknesses:**

Strengths
- The paper is well-written and well-organized, presenting a clear and compelling motivation for the study.
- The proposed overlap-based loss functions are creative and adaptable, delivering impressive results across multiple evaluation metrics and model architectures.
- The framework requires no additional training or fine-tuning, making it computationally efficient and easily adaptable to existing models.
- The experiments are comprehensive and solid, not only covering various evaluation metrics but also demonstrating the robustness of the proposed methods across different model backbones and prompt complexities.


Weaknesses
- The evaluation does not include prompts involving human subjects or complex object interactions, which limits the assessment of the method's versatility in real-world scenarios.

---

> ### Author Response · Authors · 2025-01-05
> **Answer**
>
> We sincerely thank you for your thoughtful and detailed feedback. Moreover, we deeply appreciate your recognition of the clarity and organization of our paper and your acknowledgment of the creativity and adaptability of our proposed overlap-based loss functions. Your positive feedback on our experiments' computational efficiency, robustness, and comprehensiveness is highly encouraging and valuable for improving our work.
>
> We would like to address your concerns and questions:
>
> **Issue:** The evaluation does not include prompts involving human subjects or complex object interactions, which limits the assessment of the method's versatility in real-world scenarios.
>
> **Answer:** To evaluate the versatility of our method in real-world scenarios, we extended our assessments beyond structured prompts to include diverse, naturally occurring free-style prompts extracted from the COCO dataset, which features human subjects and complex object interactions. The results of these evaluations are summarized in Table 12 of the revised manuscript.
> Additionally, we conducted new experiments during rebuttal period to assess the effectiveness of our proposed method in addressing other compositional generation challenges, including attribute binding, spatial relationships, and counting-related issues. These results, detailed in Table 8 of the revised manuscript, demonstrate the method’s robustness in handling more realistic and varied compositions.
>
> **Requested Change:** In the Appendix, the authors have presented experimental results on free-style prompts, which show promising performance. I suggest including some qualitative results as well to provide a clearer visualization of the model's effectiveness in these scenarios.
>
> **Answer:** The requested qualitative results have been incorporated into the revised manuscript and are presented in Figure 8.
>
> We sincerely appreciate your thoughtful review.

---

### Review · Reviewer_9HB8 · 2024-12-23

**Summary Of Contributions:**

Here is the extracted list of contributions:

1. Insight into the competition of entity-related tokens for attention toward specific regions via the cross-attention mechanism.
2. Proposal of four overlap-based loss functions for manipulating latent embeddings during inference:
   - Intersection over Union (IoU)
   - Center-of-Mass (CoM) distance
   - Kullback–Leibler (KL) divergence
   - Clustering
3. Introduction of training-free methods to improve compositional alignment.
4. Extensive experimentation on diverse prompts to evaluate performance.
5. Demonstration of superior performance over previous approaches across metrics such as:
   - Visual question answering
   - Captioning score
   - CLIP similarity
   - Human evaluation.

**Audience:**

Yes

**Claims And Evidence:**

Yes

**Requested Changes:**

Indicated in weaknesses.

**Strengths And Weaknesses:**

Strengths:
1. The hypothesis of the paper is pretty interesting and potentially impactful.
1. The writing is clear and captures all major facets.
1. The paper evaluates the image generation across multiple metrics like CLIP similarity, VQA, captioning, human eval, etc.
1. The paper evaluates the image generation on multiple datasets, including COCO-Comp, T2I-CompBench, and HRS-Bench etc
1. The paper evaluates their hypothesis on multiple backbones, including SD-1.4, SD-2, SD-XL

Weaknesses:
1. Within the paradigm of composition, the paper studies only "and" type relationships, and that too for only two entities.
1. Not clear if the three causes of the entity missing problems ((1) attention intensity, (2) attention spread, and (3) attention overlap), are mutually exclusive or even collectively exhaustive.
1. The hypothesis is interesting and the literature review also is quite extensive. To make the paper stronger, I would suggest that the authors add evaluations of their proposed methods with datasets from other entity related problems observed in diffusion, such as incorrect spatial relations, numeracy challenges, incorrect spatial relationships, etc.
1. Missing citation for "This problem typically arises when the model does not allocate sufficient attention to certain entities during the generation process, leading to their partial or total absence in the final output."

---

> ### Author Response · Authors · 2025-01-05
> **First Part of the Answer**
>
> We sincerely thank for your detailed and thoughtful review and deeply appreciate your acknowledgment of the clarity and comprehensiveness of our writing, as well as your recognition of our extensive evaluations. Specifically, your positive feedback on the hypothesis and its potential impact inspires us to continue refining and expanding this line of research. Thank you for your careful and constructive review.
>
> We would like to address your concerns and questions:
>
> **Issue:** Within the paradigm of composition, the paper studies only "and" type relationships, and that too for only two entities.
>
> **Answer:** As demonstrated in **Section D of the Appendix**, we have conducted assessments that extend beyond 'and' relationships between two entities. Specifically:
>
> 1. **Three-Entity Prompts:** We expanded our assessments to include three-entity prompts, with the results provided in Table 3 and Table 14 of the revised manuscript.
> 2. **Four-Entity Prompts:** Our evaluations also encompass four-entity prompts, as detailed in Table 11 of the revised manuscript.
> 3. **Free-Style Prompts:** Beyond structured 'and' prompts, we tested our method on diverse, naturally occurring free-style prompts extracted from the COCO dataset, with results presented in Table 12 of the revised manuscript.
>
> These additional evaluations underscore that our approach is not confined to two-entity compositions but is applicable to broader multi-entity scenarios with varying prompt structures. We believe these results enhance the evidence for the generalizability of our method beyond the specific case highlighted in the review.
>
> **Issue:** Not clear if the three causes of the entity missing problems ((1) attention intensity, (2) attention spread, and (3) attention overlap) are mutually exclusive or even collectively exhaustive.
>
> **Answer:** We would like to clarify three key points regarding the relationship between attention intensity, attention spread, and attention overlap:
>
> 1. **Correlation Analysis:** As illustrated in Figure 5 (right), we analyzed the correlation among these three factors. The results reveal a strong correlation between the three overlap-based metrics (CoM Distance, IoU, KL Divergence), whereas the correlation between overlap-based metrics and non-overlap-based metrics (intensity and spread) is comparatively weaker.
> 2. **Evaluation of Combined Effects:** In Section D.4 (Table 17 of the revised manuscript) of the Appendix, we assessed the combined effects of these factors in mitigating the entity missing issue. While all combinations demonstrate improvement compared to the baselines, the overlap-based metrics have the most significant impact in addressing the entity-missing problem.
> 3. **Clustering Viewpoint:** As discussed in the last paragraph of Section 4.3, we propose that the cross-attention mechanism during denoising steps can be analogized to a clustering process, where each key vector associated with an entity represents a cluster center, and each query vector, corresponding to the latent code at a spatial location, represents a data point in the clustering process. Within this framework, the phenomenon of entity missing is analogous to a cluster associated with an entity becoming empty. An entity may be missed either because its corresponding cluster is weak (i.e., containing few data points) from the initial step of denoising or because the cluster weakens due to overlap with other dominating clusters. Given that most real-world applications of T2I models involve prompts with multiple entities, our work focuses on reducing the likelihood of a cluster becoming empty by minimizing the overlap between clusters of different entities. One class of loss functions that can effectively reduce cluster overlap are overlap-based metrics applied to the attention maps of entities. From this perspective, while attention intensity and attention spread are somewhat related to attention overlap, they are not as strongly connected to the issue of cluster overlap as attention overlap itself.
>
> Therefore, these factors are not mutually exclusive, as they can occur simultaneously, and as previously highlighted, attention overlap plays the most critical role in addressing the issue of entity missing, particularly in real-world scenarios involving multi-entity prompts.

---

> ### Author Response · Authors · 2025-01-05
> **Second Part of the Answer**
>
> **Issue:** The hypothesis is interesting and the literature review also is quite extensive. To make the paper stronger, I would suggest that the authors add evaluations of their proposed methods with datasets from other entity related problems observed in diffusion, such as incorrect spatial relations, numeracy challenges, incorrect spatial relationships, etc.
>
> **Answer:**
> Regarding the suggestion to evaluate our method on additional entity-related challenges, we would like to clarify that we have extended our evaluations to encompass tasks involving spatial relationships, numeracy-related (counting) challenges, and attribute binding issues, as detailed in Table 8 of the revised manuscript.
>
> These findings demonstrate that solely improving entity preservation can also effectively enhance overall compositional alignment in text-to-image generation. More specifically, using the CoM distance metric, improvements of approximately **14%** and **7%** are achieved for the attribute binding and spatial relationship cases, respectively. Furthermore, a reduction of **0.65** in the mean absolute error (MAE) metric is observed for the counting problem.
>
> **Issue:** Missing citation for "This problem typically arises when the model does not allocate sufficient attention to certain entities during the generation process, leading to their partial or total absence in the final output."
>
> **Answer:** We have updated the paper and added the appropriate citation to support this statement.
>
> Again, we sincerely appreciate your valuable suggestions and comments.

---

### Decision · Action_Editor_Ts8v · 2025-02-10

**Recommendation:** Accept with minor revision

**Comment:**

The entity missing is an important problem in text-to-image diffusion models. This paper investigates the problem and concludes that attention overlap is the main reason with an extensive experiments. The reviewers initially raised some concerns such as the generality of the proposed method, the reasonability of evalluations, and related works, etc. The authors address most of the concerns by conducting further experiments and add more clarifications, which are appreciated by the reviewers. The paper receives unanimous positive support.

**Audience:**

The paper could benefit researchers of diffusion models.

**Claims And Evidence:**

This paper proposes three possible reasons of entity missing in T2I diffusion models. To verify the assumptions, an extensive experiments are conducted, which concludes that the attention overlap is the main reason for the entity missing issue. Overall, the claims are supported by sufficient evidence, and the results are fairly convincing.